



# The influence of hyporheic fluxes on regional groundwater discharge zones

Brian Babak Mojarrad[1], Anders Wörman[1], Joakim Riml[1], Shulan Xu[2]

[1]Department of Sustainable Development, Environmental Science and Engineering, KTH Royal Institute of Technology, Stockholm, 100 44, Sweden
[2]Xu Environmental Consulting AB, Stockholm, 168 61, Sweden

*Correspondence to*: Brian B. Mojarrad (mojarrad@kth.se)

**Abstract.** The importance of hyporheic water fluxes induced by hydromorphologic processes at the streambed scale and their consequential effects on stream ecohydrology have recently received much attention. However, the role of hyporheic water fluxes in regional groundwater discharge is still not entirely understood. Streambed-induced flows not only affect mass and heat transport in streams but are also important for the retention of solute contamination originating from deep in the subsurface, such as naturally occurring solutes as well as leakage from the future geological disposal of nuclear waste. Here, we applied a multiscale modeling approach to investigate the effect of hyporheic fluxes on regional groundwater discharge in the Krycklan catchment, located in a boreal landscape in Sweden. Regional groundwater modeling was conducted using COMSOL Multiphysics constrained by observed or modeled representations of the catchment infiltration and geological properties, reflecting heterogeneities within the subsurface domain. Furthermore, streambed-scale modeling was performed using an exact spectral solution of the hydraulic head applicable to streaming water over a fluctuating streambed topography. By comparing the flow fields of watershed-scale groundwater discharge with and without consideration of streambed-induced hyporheic flows, we found that the flow trajectories and the distribution of the travel times of groundwater were substantially influenced by the presence of hyporheic fluxes near the streambed surface. One implication of hyporheic flows is that the groundwater flow paths contract near the streambed interface, thus fragmenting the coherent areas of groundwater upwelling and resulting in narrow "pinholes" of groundwater discharge points.

## 1 Introduction

Landscape topography produces a wide distribution of groundwater flow paths that control the fate of subsurface contamination and the transport of heat and solutes into discharge zones. Large-scale flow is of great concern for radioactive waste management programs for which leakage scenarios and dose impact assessments are based upon understanding the regional groundwater flow field. One recognized option for storing high-level radioactive waste (HLRW) is to isolate the disposal repository in a deep, stable geological formation several hundred meters beneath the topographic surface (Pusch et al., 2011). However, due to the importance of human health and environmental safety, investigations of the long-term consequences of potential leakage from deep HLRW repositories are needed (Hartley and Joyce, 2013; Helton et al., 2014), e.g., investigations





using biosphere models (Hjerpe and Broed, 2010; Saetre, 2015). Analyses of radionuclide turnover in the biosphere and geosphere are generally conducted using compartment models in which radioactive contamination leakage is transferred among a number of subsurface compartments and is subsequently released in groundwater discharge in surface waters in locations where the uptake of radioactivity by biota can occur (Avila et al., 2010). Regional groundwater flow is one of the

main processes affecting radionuclide transfer among various geosphere compartments. A critical compartment in these models is the geosphere-biosphere interface (GBI), which refers to Quaternary deposits wherein water flows are generally slower than in sparse fracture networks in crystalline bedrock. In Scandinavia, the relatively thin Quaternary deposits located near streambed surfaces are considered a critical compartment in the biosphere dose model due to the (a) longer radionuclide travel time in the soil layer than in stream water and (b) accumulation due to the adsorption of radionuclides in soil that can

endanger future agricultural products (Berglund et al., 2009; Kautsky et al., 2013). Therefore, groundwater flow and the properties influencing the magnitude and direction of the water flow in Quaternary deposits are of key importance for assessing the long-term safety of radioactive deposits. In particular, streambed sediments play a predominant role in stream water quality, as conveyors and regulators of radioactivity to ecosystems and, hence, strongly influence the health of humans and other biota. Early field investigations used natural trace elements, such as $^{222}$Rn, and temperature to separate groundwater flows from

shallower hyporheic fluxes within streamed sediments. Such analyses allow the estimation of the locations of deep groundwater discharge zones and their contributions to the surface water flow (Dimova and Burnett, 2011; Cranswick et al., 2014). Previous research investigated the radionuclide transport time in bedrock at different radionuclide origin depths (Wörman et al., 2007a; Selroos and Painter, 2012) but did not consider the detailed hydraulic flow behavior of discharge zones located immediately beneath streams. Radionuclide transport in Quaternary deposits has been mainly investigated near the

ground surface without consideration of the heterogeneity in the hydraulic conductivity of stream-bed soil strata or the role of local hyporheic fluxes beneath streams. Values found in the literature indicate that local hyporheic flow velocities are on the order of 20,000 – 200,000 mm/y (Seiler and Lindner, 1995; Magri et al., 2009; Mojarrad et al., 2019b), which is much higher than the regional groundwater recharge and discharge rates that are constrained by the available infiltration and can be on the order of a few hundred to thousands of mm/y (Batelaan et al., 2003; Caruso et al., 2016). This suggests that local hyporheic

flows can have dominating effects in deep groundwater discharge zones and hence on solutes and heat transport associated with the flow of deep groundwater into streams. In particular, the residence times of radionuclides in streambeds and the sizes of regional groundwater discharge areas can be affected by hyporheic flows, which control radionuclide spreading in stream ecosystems and radiological dose entries for humans and other biota (Xu et al., 2008).

The aim of the present study is to investigate the principle effects of hyporheic flows on the discharge of groundwater

originating from depths much larger than hyporheic flow depths. The general approach is, thus, to decouple the subsurface flow processes and analyze the catchment-scale groundwater flow field and its associated discharge zones with and without consideration of hyporheic flows. We hypothesize that hyporheic fluxes substantially influence deep groundwater discharge into streambeds and that hyporheic fluxes, in combination with other factors (such as landscape topography, geology, and climate), controls groundwater flows at the watershed scale. In the present study, a multiscale modeling framework was



established to represent the spatial scales of both the geomorphology of a region and its subsurface flow field; this model was
then applied to a boreal watershed in Sweden. The multiscale modeling framework used in this study was first introduced by
Mojarrad et al. (2019b), but the methodology presented here has been substantially modified and improved to facilitate the
analyses of the effects of streambed-induced hyporheic fluxes on deeper groundwater discharge. The major methodological
differences of the present study are due to (a) the application of more observational constraints to catchment- and hyporheic-
scale modeling (depth decay hydraulic conductivity, infiltration constraints at the topography surface, different hydraulic
conductivity values for the hyporheic zone), (b) the performance of the particle tracing analysis in the catchment-scale model
for evaluating the deep groundwater flows, (c) the layering of the catchment-scale model into three domains representing
sediments, Quaternary deposits, and bedrock, and (d) the application of a comprehensive study on the evaluation of hyporheic
flow fields using Monte Carlo simulations. Overall, the improved modeling methodology was a prerequisite for the analyses
of the effect of streambed-induced hyporheic fluxes on deeper groundwater discharge.

## 2 Methodology

### 2.1 Modeling Framework

A steady-state multiscale modeling framework was applied in this study to cover the spectrum of scales affecting the
interactions between the groundwater and surface water that occur within streambed sediments. Similar to the work of
Mojarrad et al. (2019b), the flow problem was addressed at two spatial scales: the catchment scale and the streambed scale. At
the catchment scale, the model was constrained by data representing the landscape topography, geological heterogeneity of
both the Quaternary deposits and bedrock, stream network extension and the mean annual infiltration rate controlling the
overall circulation intensity of the groundwater (see section 2.2 for data description). The application of Darcy's law for steady-
state subsurface flows within a saturated porous media results in the 3D steady-state groundwater flow equation, described as
follows:

$$\frac{\partial}{\partial x}\left(-K_x \frac{\partial H}{\partial x}\right) + \frac{\partial}{\partial y}\left(-K_y \frac{\partial H}{\partial y}\right) + \frac{\partial}{\partial z}\left(-K_z \frac{\partial H}{\partial z}\right) = 0 \qquad (1)$$

where H (m) represents the total hydraulic head, $H = z_e + p/(\rho \times g)$; $z_e$ (m) is the elevation; p (Pa) is the pressure; g (m/s$^2$) is the
acceleration due to gravity; $\rho$ (kg/m$^3$) is the water density; and K (m/s) is the hydraulic conductivity. Moreover, x (m), y (m),
and z (m) are the spatial directions, among which x and y lie on the horizontal surface and z represents the vertical spatial
direction (positive upward). Since equation (1) is a linear function, the superposition principle can be used to separate the total
hydraulic head, H(x,y,z) (m), into individual terms operating on different spatial scales, as follows:

$$H(x, y, z) = H_S(x, y, z) + H_C(x, y, z) \qquad (2)$$



where $H_S$ (m) is the streambed-induced hydraulic head fluctuation and $H_C$ (m) is the catchment-scale hydraulic head. The catchment-scale hydraulic head reflects the trend and fluctuation in the catchment-scale water surface elevation, whereas the

streambed-induced hydraulic head can be regarded as a local detrended perturbation of the water surface along the streams and lakes. In this study, the catchment-scale modeling was conducted in COMSOL Multiphysics® software (section 2.3). However, an exact solution was used to represent $H_S$ for streambed-induced flow modeling (section 2.4). The mean of the absolute value of Darcy's vertical velocity of the catchment scale model at z=z´ is evaluated as follows:

$$\langle |W_C| \rangle = \frac{1}{A_z} \int_{A_z} (|W(x, y, z = z´)|) \ dA_z \tag{3}$$

where $W_C$ (m/s) is Darcy's velocity in the z direction, $A_z$ (m²) is the surface area, z´ (m) is any arbitrary depth, and d is the differential operator. Darcy's velocity is defined as the flow velocity averaged over the entire cross-sectional area of a porous medium. However, the seepage velocity (also called the pore water velocity) is defined as $q_{seepage} = \frac{q}{n}$. Thus, $q_{seepage}$ is the flow velocity divided by the porosity of the media, n (-), and is representative of the transport time of inert water in groundwater. Solute retardation was added to estimate the solute transport time in different subsurface strata. In this study, the

retardation of the radionuclide transport time in different layers was evaluated for $^{135}$Cs using $t_{solutes} = (1 + R) \times t_{water}$, where t (s) is the transport time and R (-) is the retardation factor for each subsurface stratum due to the sorption and diffusion processes. It should be noted that $t_{water}$ corresponds to the travel time of water, estimated by the seepage velocity. The values used for the retardation factor in rock fractures as well as in Quaternary deposits and sediments were $R_{rock} = 10$ and $R_{QD, sediment} = 500$ (Wörman et al., 2004).

## 2.2 Empirical and Observational Data

The study was performed using data from the Krycklan catchment, which is located in northern Sweden, near the city of Umeå (64°14´N, 19°46´E). Krycklan is a research catchment with a 68-km² area and extensive field infrastructure for monitoring hydrology, water quality, stream biodiversity, and climatology (Laudon et al., 2013). The catchment consists of 15 subcatchments that are instrumented for discharge measurements (Laudon et al., 2007). The measured stream discharges along

with the physical characteristics of the stream network were used to quantify the streambed-induced flow field (section 2.4). High-resolution topographical digital elevation data exist for the whole catchment; in these data, the horizontal and vertical resolutions are 2 m and 1 cm, respectively. The catchment elevation is in the range of 117-405 m above sea level. Bedrock surface elevation data with a horizontal resolution of 10 m are provided by ©Sveriges geologiska undersökning (SGU). The thickness of the Quaternary deposits is quantified by subtracting the bedrock surface elevation from the topographical digital

elevation data. The soil type data are made available at the Kryckan catchment website (https://www.slu.se/krycklan), and the dominant soil type mostly includes tills in the highland regions that are gradually replaced with deeper, sandy sediments (up to 40-m in depth) towards the catchment outlet (Lyon et al., 2011). Seven different soil types exist in the Krycklan catchment (Figure S1, Supporting Information), and the hydraulic conductivity of each soil type was obtained by Sterte et al. (2018) to





reflect heterogeneities in the Quaternary deposits. The streambed sediment was defined as the top 5 m of soil beneath the
stream network (with a 5-m width) in which a hydraulic conductivity of $9.13\times10^{-4}$ (m/s) was considered at the streambed
interface, as obtained from the experiment of Morén et al. (2017) (section 2.3). In addition, hydraulic conductivity was
considered to exponentially decay with depth. The mean annual precipitation and temperature (from 1981 to 2007) were 614
mm and 1.7°C, respectively, and half of the precipitation fell as snow within this time period (Haei et al., 2010). The annual
mean run-off was set as the infiltration from the recorded precipitation. Steady state modeling allowed the use of infiltration
as the catchment-scale constraint so that the mean value of the vertical flux at the top surface of the catchment-scale model did
not exceed the mean infiltration value. This constraint was fulfilled by using the variable resolution of the topographic elevation
data (section 2.3).

**2.3 Regional-Scale Groundwater Flow Numerical Modeling**

The catchment-scale numerical model was conducted in COMSOL Multiphysics®. The whole domain was stratified into three
horizontal layers representing the streambed sediments, Quaternary deposits, and bedrock. Here, the streambed sediment was
defined as the top five meters of Quaternary deposits, while the depth of the Quaternary deposits (sediments and underlying
soil strata) was obtained through analysis of the soil depth data. Previous studies have shown that hydraulic conductivity decays
with depth for most geological features (Ingebritsen and Manning, 1999; Saar and Manga, 2004; Jiang et al., 2009; Grant et
al., 2014; Ameli et al., 2015). Hence, the hydraulic conductivity of each individual layer is described according to an
exponential function that reflects a decreasing decay with depth in all media, as follows:

$$K_i(z) = K_{(S\ or\ QD\ or\ B),top}\ e^{\frac{(z-z_{top,i})}{\delta_i}} \tag{4}$$

where the subscript $i$ represents the different layers; $K_{S,top}$ (m/s), $K_{QD,top}$ (m/s), and $K_{B,top}$ (m/s) are the hydraulic conductivities
at the top surfaces of the streambed sediment (S), Quaternary deposit (QD), and bedrock (B) layers, respectively; $z_{top,i}$ (m) is
the elevation at the top surface of the corresponding layer; and $\delta$ (m) is the skin depth. Saar and Manga (2004) indicated that
the skin depth varies in the range of 200-300 m for both Quaternary deposits and bedrock. In the present study, we considered
$\delta = 250$ m for the Quaternary deposit and bedrock layers. Furthermore, $1/\delta$ is an empirical decay coefficient for streambed
sediments that can be determined experimentally (Marklund and Wörman, 2011). In this study, the decay coefficient of the
streambed sediment layer was calculated based on the depth-varying hydraulic conductivity observations reported by Morén
et al. (2017) (Figure S2, supporting information).
Additionally, the heterogeneity in the hydraulic conductivity of the Quaternary deposits ($K_{QD}$) was considered using a soil map
and the associated hydraulic conductivity patterns over the Krycklan catchment (Sterte et al., 2018). The hydraulic conductivity
of the top surface of the bedrock layer was chosen based on the values provided by Ericsson et al. (2006). The porosity of the
bedrock layer was assumed to be 0.001, whereas the porosities of the Quaternary deposits and sediment layer were assumed
to be 0.2 (Figure 1).





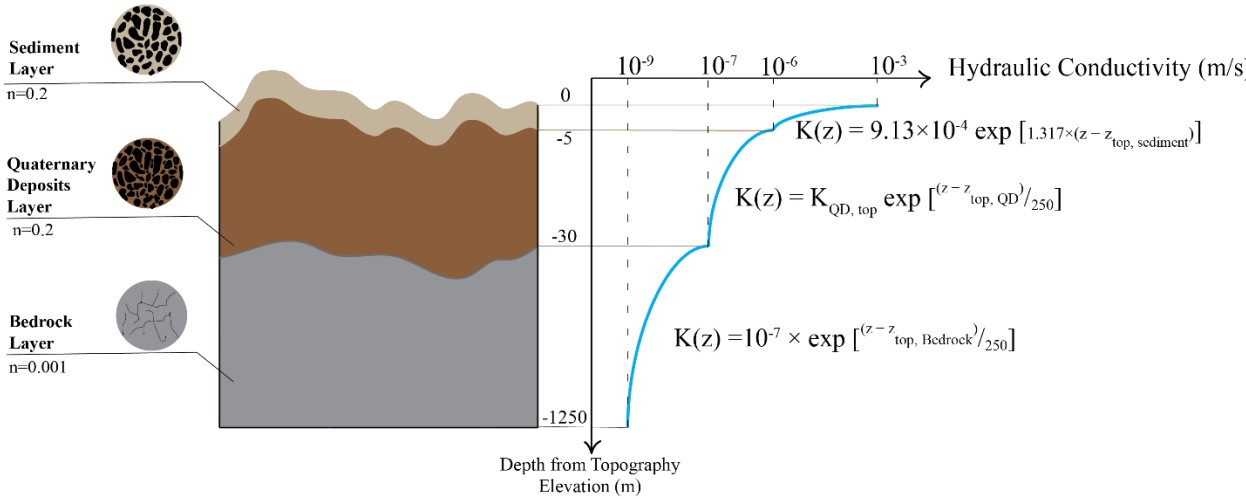

**Figure 1.** Schematic sketch of the considered porosity, n, and depth-decaying hydraulic conductivity, K, in the three different layers of the model. Note that the axes are not to scale.

A nonuniform mesh was used for the flow calculation of each layer. The mesh sizes varied within the ranges of 0.1-2 m, 2-17 m, and 17-403 m for the streambed sediment, Quaternary deposit, and bedrock layers, respectively. The water table in the Krycklan catchment was classified as topography-controlled, which reflects that the water table is a subdued, smoothed replica of the elevation of the landscape (Mojarrad et al., 2019b). However, the correlation between the fluctuating topography and groundwater level may spatially vary due to the impact of infiltration. In particular, topography has a controlling local impact on the water level where surface water resources exist, whereas infiltration plays a major role in terrain with high local elevations (Desbarats et al., 2002; Sanford, 2002). Hence, the realistic condition is to set topography-controlled boundary conditions (i.e., Dirichlet) at lakes and streams while setting a recharge-controlled boundary condition (i.e., Neumann) for the rest of the terrain. Previous studies indicated that the resolution of the chosen digital elevation model (DEM) strongly affects the vertical groundwater flux at the topography surface, i.e., the infiltration or groundwater recharge (Marklund and Wörman, 2011; Wang et al., 2018). Therefore, the top boundary condition of the numerical model was aligned with the landscape topography elevation but was smoothed by changing the DEM resolution over (only) the recharge areas to constrain the recharge rate to the observed infiltration of the catchment. The applied method helps to satisfy both the Dirichlet and Neumann boundary conditions through a practical way of recognizing unknown and spatially variable infiltration. The smoothing of the groundwater table was performed by lowering the resolution of the mesh size of the identified recharge areas via an iterative mesh approach until the annual infiltration rate of the catchment was satisfied.

No flow boundary condition was assumed for the flat horizontal surface at the bottom of the model. Additionally, the total depth of the domain, $D_T$, varied spatially within the catchment, ranging from 950 to 1250 m depending on the surface elevation. In addition, the impact of the sides of the numerical model was accounted for by applying a constant hydraulic head to the





lateral surfaces of the rectangular domain ($H_C(x, y, z)|_{x=0,L_x \text{ and } y=0,L_y} = Z_B(x,y)$, where $Z_B(x,y)$ is the landscape topography elevation and $L_x$ and $L_y$ represent the lengths of the catchment in the x and y directions, respectively).

A particle tracing routine was implemented to analyze the distribution of deep groundwater flow paths. A total of 10,000 inert particles were released from 100×100 uniformly spaced grid points over a flat horizontal surface located at a depth of approximately 500 m below the minimum topography elevation and tracked over a fictitious time period of 320 million years that allowed all particles to exit the flow domain along their corresponding particle trajectories. The concept of a time period in steady-state particle tracing is strictly numerical; particle tracing uses the evaluated velocity fields from the steady-state Darcy's law to determine particle positions. In this study, the flow paths of particles that reached the surface from a depth of 500 m are considered to represent the deep groundwater flow in all subsequent analyses.

**2.4 Streambed Induced (Hyporheic) Flow**

The streambed hydraulic head was applied as a boundary condition for the hyporheic flow, i.e., the flow of surface water through streambed sediment inflow paths that re-emerges in surface water. A hyporheic flow can be decomposed into hydrostatic and dynamic components that are both induced by the streambed topography. Here, the hydrostatic head component is conceived as the water-surface elevation, whereas the dynamic head component is defined as the deviating pressure variation at the sediment-water interface that varies due to the acceleration of flowing water over bedforms (i.e., the conversion of velocity head into a pressure head). Following Mojarrad et al. (2019b) and Marklund and Wörman (2011), the streambed-induced hydraulic head with an exponentially depth-decaying hydraulic conductivity can be represented as follows:

$$H_S(x,y,z) = \sum_{j=1}^{N} \sum_{i=1}^{N} \left[ C_{damp}(\lambda_i) + \left( \frac{h_m}{\sigma_{S,B}\sqrt{2}} \right) \right] (A)_{i,j} \sin(k_i x) \cos(k_j y) \times e^{-\left( -\frac{c}{2} + \sqrt{\frac{c^2}{4} + \alpha(k_x^2 + k_y^2)} \right) z} \tag{5}$$

where N (-) refers to the number of wavelengths, $(A)_{i,j}$ (m) are amplitude coefficients determined from the local streambed topography fluctuation, $k=2\pi/\lambda$ (m⁻¹) is the wavenumber, $\lambda$ (m) is the wavelength, c (m⁻¹) is an empirical decay coefficient, $\alpha = \frac{K_x \text{ or } K_y}{K_z}$ (-) is the anisotropy ratio, $C_{damp}(\lambda)$ is the hydrostatic damping factor representing the smoothness of the water surface in comparison to the streambed surface and $\left( \frac{h_m}{\sigma_{S,B}\sqrt{2}} \right)$ is the dynamic coefficient. Furthermore, $h_m$ (m) is the amplitude corresponding to the velocity head deviation, and $\sigma_{S,B}$ (m) is the standard deviation of the streambed topography elevation. Fehlman (1985) defined an equation for estimating the amplitude of the velocity head variation for a 2D, solid, triangular bedform shape ($Z_{BM}/\lambda = 1/7$, in which $Z_{BM}$ (m) is the bedform height (Elliott and Brooks, 1997a, b)):

$$h_m = 0.28 \left( \frac{v_f^2}{2\,g} \right) \begin{cases} \left( \frac{Z_{BM}/D_w}{0.34} \right)^{3/8} & Z_{BM}/D_w \leq 0.34 \\ \left( \frac{Z_{BM}/D_w}{0.34} \right)^{3/2} & Z_{BM}/D_w \geq 0.34 \end{cases} \tag{6}$$



where $v_f$ (m/s) is the averaged flow velocity in the stream, g (m/s$^2$) is the gravitational acceleration, and $D_w$ (m) is the flow
depth. Later, Stonedahl el al. (2010) extended Fehlman's equation to be applicable for 3D sinusoidal bedforms with an
205    amplitude of $Z_{BM} = 2\sqrt{2}\sigma_{S,B}$.

Landscape topographies have been shown to follow fractal patterns, allowing a spectral representation of the head boundary
condition as well as solutions to topography-controlled groundwater circulation (Wörman et al., 2006, 2007b). The fractality
reflects a constant power law correlation between the topographic amplitude and wavelength across all scales in a real Fourier
series representing the topography elevation. This fractal power has been shown to prevail over a wide range of scales, from
210    continental scales to bedforms in streams (Wörman, et al., 2007a), suggesting that there is a possibility of generalizing ground
surface topography, such as streambed topography, over scales where observations are lacking. Previously, methods for
rescaling observed topography to smaller scales to generate streambed topography have been applied (Morén et al., 2017,
Mojarrad et al., 2019b). Here, a similar rescaling method was conducted using landscape topography with a size of 100 m ×
100 m and a resolution of 2 m × 2 m identified on a DEM of the topography surrounding the stream. These regions were
rescaled to a size of 5 m × 5 m and a resolution of 0.1 m × 0.1 m to represent the streambed topography. The details of the
rescaling process are described in Mojarrad et al. (2019b).

## 2.5 Spatial Representativity of the Streambed Induced Flow

This study recognized uncertainties in the ways in which hydrostatic and dynamic head boundary conditions were represented
using equation (5), while other models and statistical uncertainties associated with the subsurface flow modeling were not
formally analyzed. Due to the uncertainty in the hydrostatic damping factor and dynamic head coefficient, a Monte Carlo
sensitivity analysis was conducted to provide a statistically representative sampling of the possible streambed-induced flow
fields. The hydrostatic damping factor reflects the ratio of the water surface to bedform variations and ranges between 0 and
1. In addition, the streambed-induced dynamic head coefficient depends on the flow velocity, flow depth, and variation in the
bedform elevation (equation 6). Since the aim of the study is to investigate the impact of hyporheic fluxes on deep groundwater
discharge zones, the stream flow velocity and flow depth of the hyporheic dynamic head coefficient should be representative
of the deep groundwater discharge points along the stream-lake network. Hence, the catchment-scale groundwater flow
discharge points were clustered into 30 different subcatchments (based on the deep groundwater discharge locations at the
topography surface) to represent a broad range of stream flow characteristics within the study area. Then, the flow velocity
and water depth were estimated for each subcatchment based on the corresponding drainage areas and mean values of the
physical characteristics of the stream segments identified for each of the subcatchments (Figure S3 and Table S1 in the
supporting information). Consequently, the deep groundwater discharge points at the topography surface that were grouped
together were assumed to have the same flow velocity and flow depth values in the streambed-scale model.

The spatial variation in streambed topography was accounted for by considering 20 arbitrarily selected (Figure S1, Supporting
Information) and rescaled streambed topographies (see section 2.4). Although this method does not provide estimates of
streambed topography at all modeled groundwater discharge points, the method provides a broad range of possible bedform





variations within the catchment. Furthermore, the uncertainty in the quantitative analysis of the streambed-induced flow field was accounted for by conducting a Monte Carlo analysis based on 400 randomized combinations of hydrostatic damping factors (from a uniform distribution of [0-1]) with streambed topographies (from 20 rescaled streambed bedform variations). Finally, these 400 samples obtained from the Monte Carla analysis were used to calculate the hydraulic head for each of the 30 clusters of deep groundwater discharge points (reflecting 30 different stream flow velocities and water depths), and the results were used to represent the uncertainty for a given cluster. Consequently, when analyzing the whole watershed, 12,000 (= 400×30) realizations of the hydraulic head variability at the streambed scale were used. A summary of the samples applied in the analysis is presented in Table 1.

**Table 1.** Schematic Description and Summary of the Considered Samples in the Analysis

| Component | Streambed-Induced Hydraulic Head Components | | | |
|---|---|---|---|---|
| | Dynamic Head Coefficient | | | Hydrostatic Head Damping Factor |
| | $\left(\dfrac{h_m}{\sigma_{S,B}\sqrt{2}}\right)$ | | | $C_{damp}$ |
| **Principal Parameter** | $v_f$ | $D_w$ | $\sigma_{S,B}$ | |
| **Description of the Considered Samples** | 30 different subcatchments (Figure S3, Supporting Information) | | 20 different streambed topography variations (Figure S1, Supporting Information) | Random number within the 0-1 interval |
| **Included in Monte Carlo Simulation** | - | | ✓ | ✓ |
| **Number of Samples** | 30 | | 400 | |
| **Number of 5×5×5 m³ Streambed-Scale Realizations** | 30 × 400 = 12000 | | | |

## 2.6 Particle Tracing in Superimposed Flow Fields

The 12000 realizations of streambed-induced flow involved 400 streambed-scale models for each of the 30 subcatchments (i.e., flow properties). The 5×5×5-m³ cubes at the deep groundwater discharge zones contained both deep groundwater flow and "intermediate groundwater flow", identified from using particle trajectories starting in recharge areas but confined to Quaternary deposits and sediment layers (i.e., without entering the bedrock domain) (Figure 2). The velocity vector for each cube containing deep groundwater particles was extracted with a resolution of 0.1 m from the catchment-scale model. The





catchment-scale groundwater velocity field was superimposed on the corresponding streambed-scale velocity field (i.e., the corresponding deep groundwater discharge zone (each of the 1552) to the 30 subcatchments). Hence, 400×1552 superimposed flow field realizations were used in this study. Subsequently, a set of 50×50 particles with a uniform grid distance interval of 0.1 m was released at the bottom of the domain of each superimposed flow field (i.e., at a depth of 5 m). The released particles

were tracked as they migrated towards their final destinations at the domain boundaries to evaluate the impact of the streambed-induced flow field on intermediate and deep groundwater flows (Figure 2). The travel times of the individual released particles in the superimposed model were evaluated.

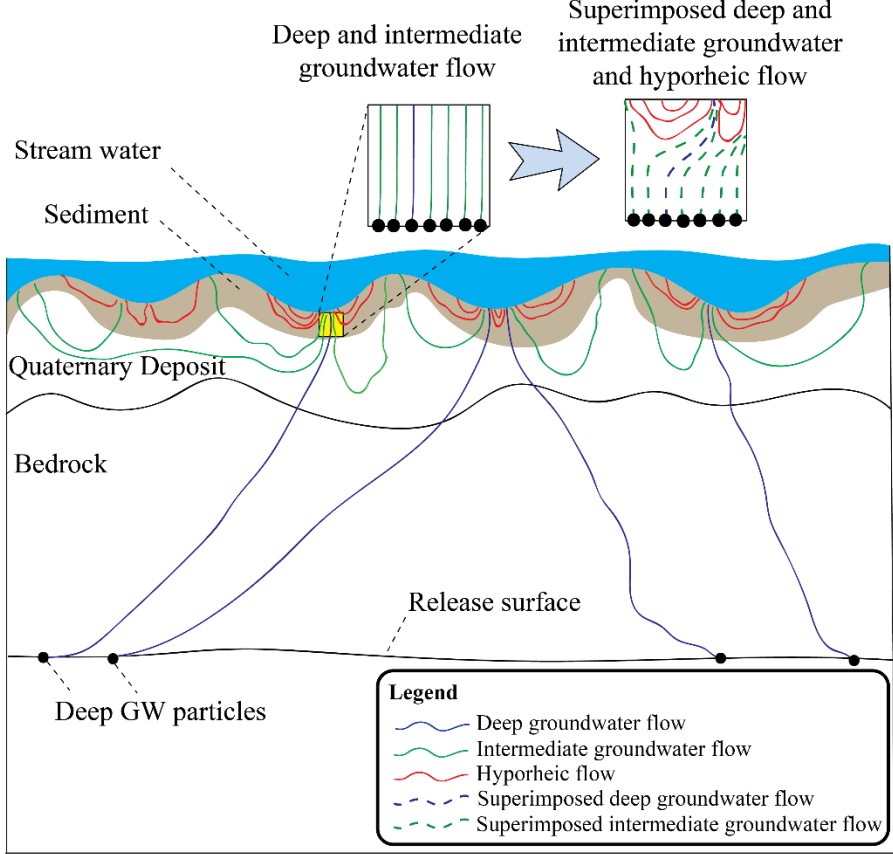

**Figure 2.** Schematic cross-sectional sketch showing the multiscale flow processes that are accounted for in the present study. The catchment-
scale model only covers the deep and intermediate groundwater flow fields (blue and green solid lines), while the streambed-scale model evaluates the hyporheic fluxes (red solid lines). The superimposed deep and intermediate flows (blue and green dashed lines) are influenced by the hyporheic fluxes. Note that the figure is not to scale.

## 2.7 Fragmentation of the Groundwater Discharge Zone

In addition to the deep groundwater travel time in the superimposed models, the analyses also covered how the hyporheic
flows affected the spatial distribution of various sizes of catchment-scale groundwater upwelling zones at streambed interfaces. These results were used to determine the fragmentation of catchment-scale groundwater flows arising at streambed interfaces,





as defined from the change in the distributions of coherent areas that only experienced the upwelling of catchment-scale groundwater flow. The changes in coherent discharge areas were determined by superimposing and not superimposing the hyporheic flows on the catchment-scale groundwater flows. In this study, a coherent area was defined as an area in which the

entire flow reflected only the catchment-scale upward groundwater flow. Numerically, coherent upwelling areas were evaluated at the top surfaces of the streambed-scale and catchment-scale models using an orthogonal mesh with resolutions of $0.1\times0.1$ m$^2$ and $5\times5$ m$^2$, respectively, wherein the flow velocity values were considered only in the orthogonal directions.

## 3 Results

### 3.1 Groundwater Flow Field

The catchment-scale groundwater results showed that the groundwater fluxes at depths shallower than approximately 70 m were substantially influenced by the resolution of the water table DEM. By successively decreasing the resolution in groundwater recharge areas, small-scale topographic features were smoothed, resulting in decreased infiltration at the upper boundary. The mean value of the absolute vertical velocity of the catchment-scale groundwater flow, $\langle |W_C| \rangle$, was used to evaluate the impact of the DEM resolution on the groundwater flow intensity. The results indicated that using a 2-m-resolution

DEM led to a flux of approximately 3000 mm/year for $\langle |W_C| \rangle$ at the water table (z = 0 m), which was higher than the average annual precipitation in the study domain (Haei et al., 2010). Successively revising the resolution of the topographical data to 70 m (only for downwelling regions) substantially influenced the vertical flux profile; $\langle |W_C| \rangle$ was found to be 500 mm/year at the topography surface at the 70-m resolution. Since this value corresponded to the approximate infiltration (i.e., the annual mean runoff from precipitation) of this area, it was kept as a plausible model setup reflecting the intensity of groundwater

circulation and, hence, the magnitudes of the discharge velocities. The results highlighted the fact that the applied resolutions of the topographic data influenced the vertical flux to a depth of 70 m but did not significantly influence the flux below this depth (Figure 3).



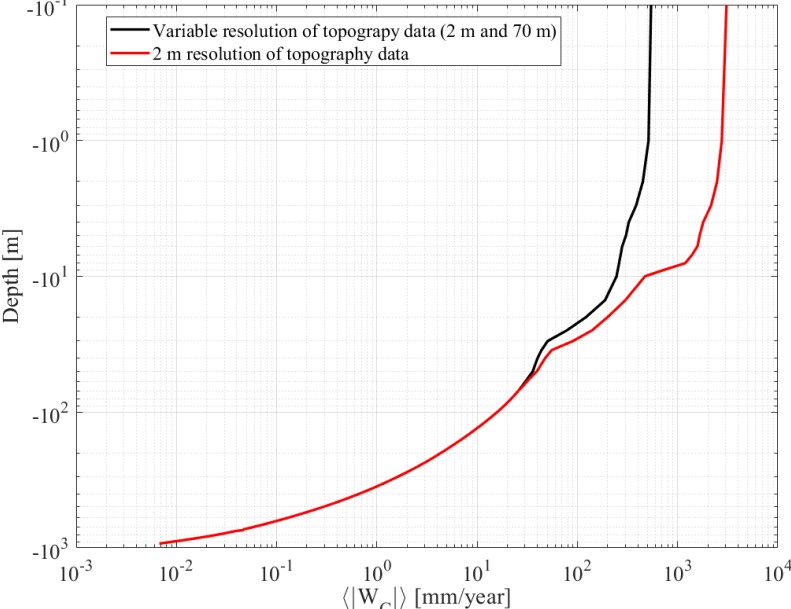

**Figure 3.** Mean values of the absolute vertical velocity of the catchment-scale model, $\langle|W_C|\rangle$, in depth. The results concern the value of $\langle|W_C|\rangle$ using a 2-m resolution DEM (red line) and a variable-resolution DEM such that the upwelling and downwelling regions had topography data resolutions of 2 m and 70 m, respectively (black line).

The discharge points of the particles released at 500-m depths were evaluated through a particle tracing method in the catchment-scale model. The particle statistics show that 4434 (=2743+1691) of the released particles had initial positive (upward) vertical velocities, and 2743 particles reached the top surface of the model. A schematic sketch showing the details of the particle tracing statistics is presented in Figure S4 in the supporting information. In addition, 1552 discharge points at the topography surface were located within the catchment area, while the rest of the particles reached the top surface (i.e., 1191 particles) outside of the catchment boundaries (Figure 4). The discharge locations for the deep groundwater particles with initial positive vertical velocities at a depth of 500 m revealed that the majority of the particles reached the surface at lowland areas along the stream network and in lakes. Eleven lakes exist in the Krycklan catchment, and the results indicated that 162 particles (or 10%) representing deep groundwater discharge points reached the lakebed surface.



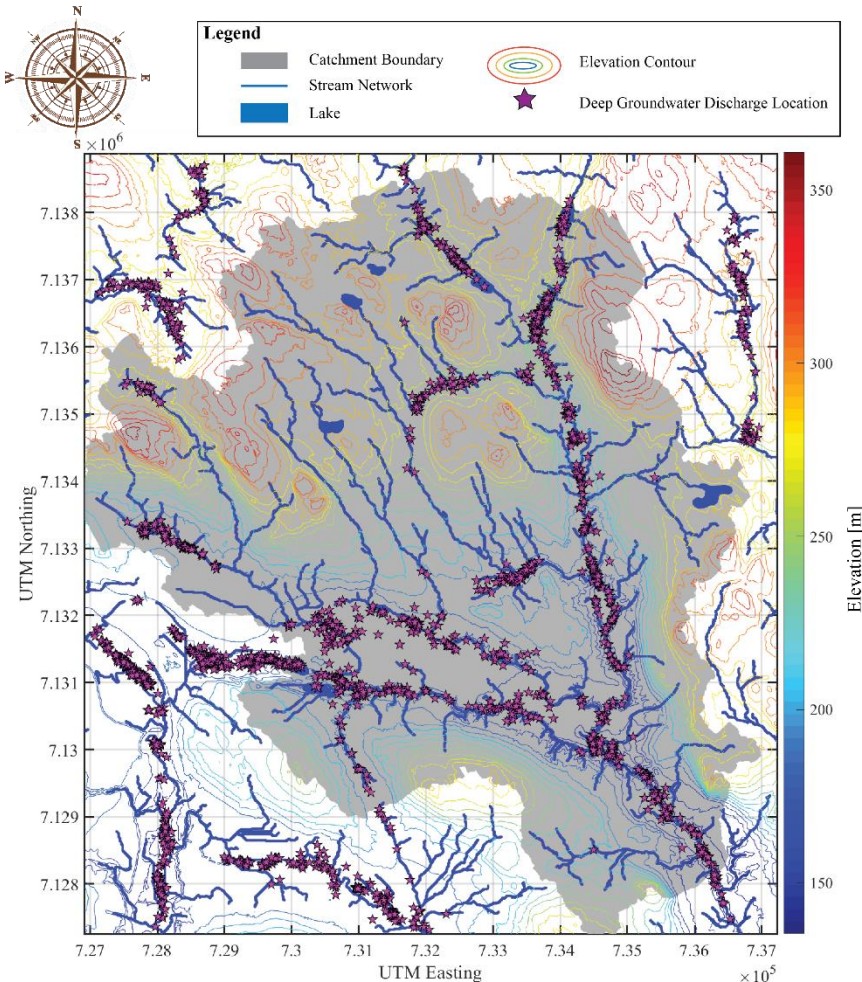

**Figure 4.** Discharge locations of particles released from a depth of 500 m from the minimum topography elevation of the catchment. The discharge locations are represented by purple stars. In addition, the topography elevation range is presented with contours in which the colors range from blue to red as the elevation increases. The stream network and the catchment boundary are represented by the blue line and gray zone, respectively.

The catchment-scale modeling results indicated that the travel times of the inert particles (quantified using the seepage velocity) that reached the top surface of the model substantially differed among different layers (i.e., bedrock, Quaternary deposit, and sediment layers) (Figure 5). This result was due not only to the different porosities, hydraulic conductivities and depths of these layers but also to a successive acceleration of the flow towards the top surface associated with the hierarchical structure of groundwater flow cells. The results indicated that the travel times for particles released at a 500-m depth were longest in the bedrock domain (in the range of 95 to 40,000 years), while the particles' travel times corresponding to the Quaternary deposit domain were substantially shorter (in the range of 3 to 60 years). The travel times of particles within the sediment layer were even shorter than those in the other layers and varied between 1-30 years. The median travel times of the particles in the bedrock, QD, and sediment layers were approximately 700, 11, and 6 years, respectively. The inclusion of radionuclide



retardation in the subsurface strata substantially increased the travel time, especially in the Quaternary deposit and sediment
layers. In particular, the median transport times of $^{135}$Cs in the different layers were all found to be on the order of $10^3$ years
(i.e., the bedrock, Quaternary deposit, and sediment layers).

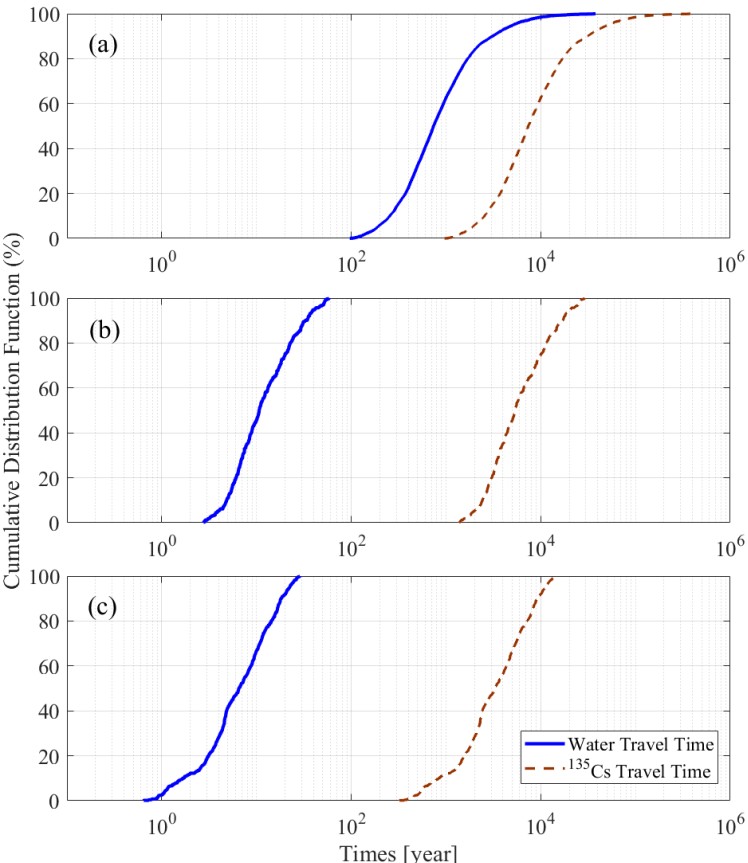

**Figure 5.** Cumulative distribution function plots of the water travel time (solid blue line) and $^{135}$Cs transport time (dashed red line) in the (a)
bedrock, (b) Quaternary deposit, and (c) sediment domains. It should be noted that the results correspond to particles with initial positive
vertical velocities that were released from a depth of 500 m from the minimum topography elevation and reached the top surface.

### 3.2 Streambed-Induced (Hyporheic) Flow Field

The streambed-scale flow fields were quantified for 12,000 realizations (i.e., 30 subcatchments representing the local stream
flow velocities and depths and 400 samples obtained by a Monte Carlo simulation with random combinations of streambed
topography variations and hydrostatic damping factors). The results demonstrated that the mean value of the absolute vertical
velocity at the streambed interface, $\langle |W_S| \rangle$, always had a median value in the range of $3\times10^{-6}$ - $9\times10^{-6}$ (m/s) (Figure S5 in the
supporting information). The dynamic coefficient varied with variations in the topography elevation as well as variations in
the stream flow characteristics within the catchment. The contributions of the hydrostatic head and dynamic head to the total
streambed hydraulic head were evaluated by comparing the results obtained through the consideration of only the hydrostatic
head damping factor or dynamic head coefficient (Figure 6 and Table S2 in the supporting information). The results showed



that the Froude number plays a major role in the relative contribution of the dynamic head coefficient; the higher the Froude number is, the larger the dynamic head contribution is (Figure 6a). The results of the Monte Carlo simulation for the study domain revealed that the contribution of the dynamic head component becomes higher than that of the hydrostatic head component when the Froude number is greater than approximately 0.450. However, most of the samples in the applied Monte Carlo simulation had Froude numbers that were lower or even much lower than 0.45, at which the hydrostatic head component dominated the total streambed-scale hydraulic head (Figure 6c).

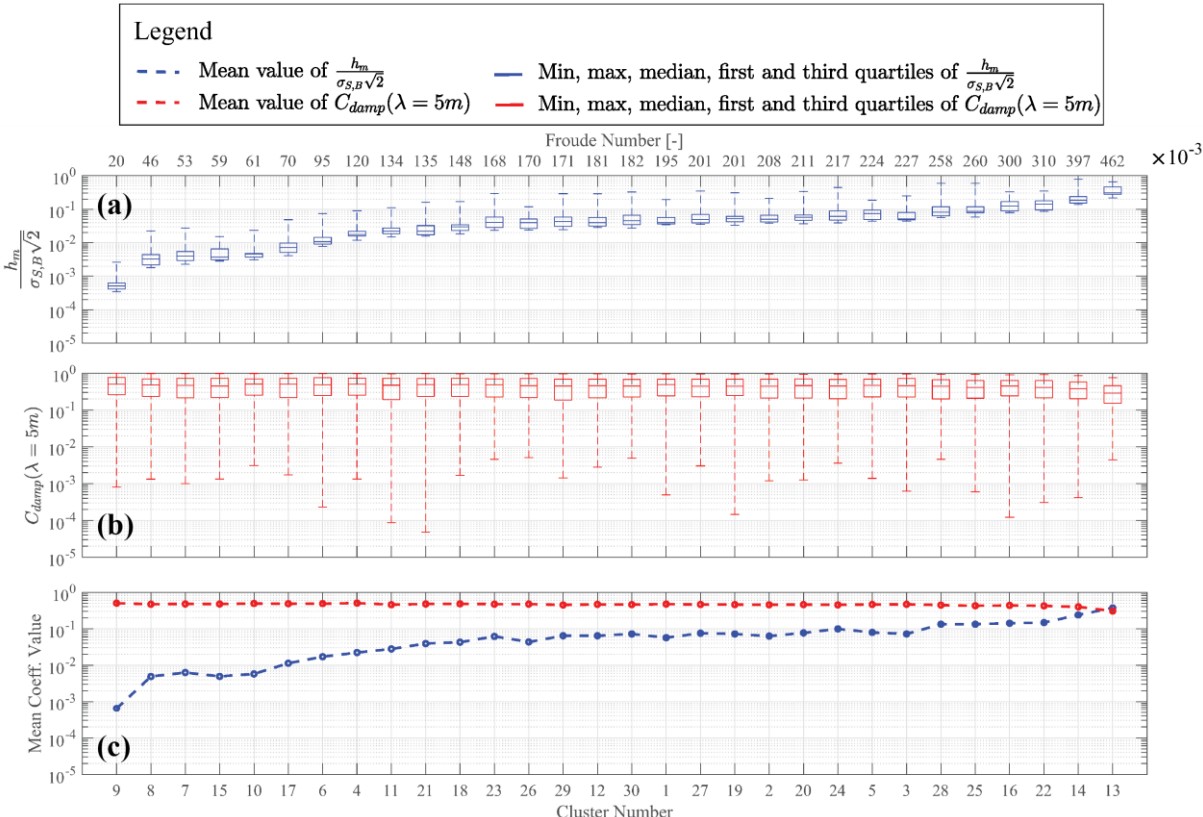

**Figure 6.** Streambed-induced hydraulic head factors that were used in the Monte Carlo simulation for different areas in different parts of the study catchment (see Figure S3 in the supporting information for the drainage area labels) sorted by the Froude number values of the flow properties in various subcatchments: **(a)** the distribution of the dynamic factor for each drainage area, **(b)** the distribution of the hydrostatic damping factor for each drainage area, and **(c)** the mean dynamic coefficient (blue color) and hydrostatic damping factor (red color) values for each drainage area.

### 3.3 The Impact of Streambed-Induced Flow on the Discharge of Regional Groundwater

Three metrics were selected to assess the impacts of streambed-induced fluxes on the groundwater flow field: (i) the travel time of groundwater flow in the streambed sediments, (ii) the groundwater direction near the bed surface, and (iii) the fragmentation of coherent groundwater upwelling areas at the sediment bed. The results revealed that the travel time of the discharging groundwater near the topography bed surface substantially depends on the penetration depth of the groundwater



flow path. The analysis of the groundwater flow field (in the absence of streambed-induced flow field) showed that intermediate flow paths that circulated only in Quaternary deposits had shorter travel times within the sediment layer than

those of deep groundwater flow emanating from a 500-m depth (Figure 7, solid lines). In particular, the travel time of the intermediate groundwater flow was only approximately $\frac{1}{3}$ of the travel time of the deep groundwater flow within the sediment layer. In addition, the superimposed flow field revealed that the streambed-induced hyporheic flow substantially decreased the travel time of the discharging groundwater within the sediment layer (Figure 7, dashed lines). This hyporheic-induced amplification of the groundwater seepage velocity was stronger for flow paths with lower seepage velocities, i.e.,

longer travel times, highlighting the strong influence of the streambed-induced fluxes on deep groundwater flow.

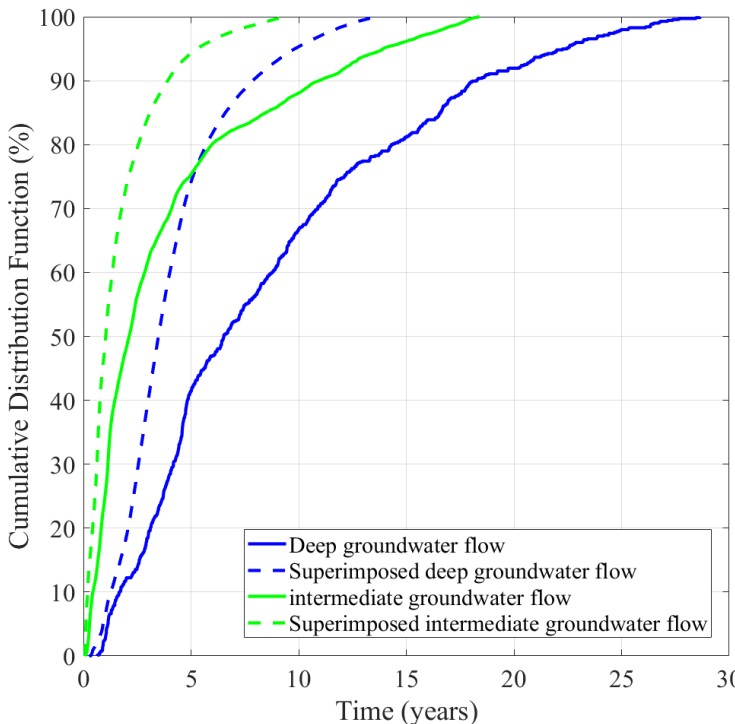

**Figure 7.** Cumulative distribution function plot of the intermediate (green color) and deep (blue color) travel times (using seepage velocity) of the groundwater flow within the sediment layer (i.e., 5×5×5 m cubes); the solid lines represent the catchment-scale results (without the streambed-induced flow influence), whereas the dashed lines represent the superimposed results (the catchment-scale flow superimposed

with the streambed-scale induced flux). Description of the different flow types are presented in Figure 2.

The groundwater flow directions near the streambed surface were substantially influenced by the streambed-induced fluxes, the hyporheic flow. Figure 8 shows the divergence of the upward groundwater flow fields from their trajectory paths due to the presence of hyporheic flow fields in two arbitrary superimposed flow fields representing a dynamic head coefficient of 0.08 and hydrostatic damping factor, $C_{damp}$ ($\lambda$=5 m), values of 0.73 (Figure 8a,b) and 0.13 (Figure 8c,d). The depth-decaying

hydraulic conductivity in the streambed-scale models controls the depth at which streambed-induced flows start to impact the groundwater flow field. Figures 8a and 8b show that the influence of hyporheic flows on the groundwater discharge begins at depths between 2.5 and 3.5 m from the topography surface, while the impact is stronger at shallower depths in Figures 8c and



8d. Due to the impact of the streambed-induced flow field, the vertical upwelling of the groundwater is separated (or fragmented) into a few detached small areas of groundwater discharge at the streambed interface. Figure 8a shows the stronger impact of the streambed-induced flow field on the upward groundwater flow compared with that shown in Figure 8c, in which smaller upwelling areas result at the streambed interface (Figure 8d).

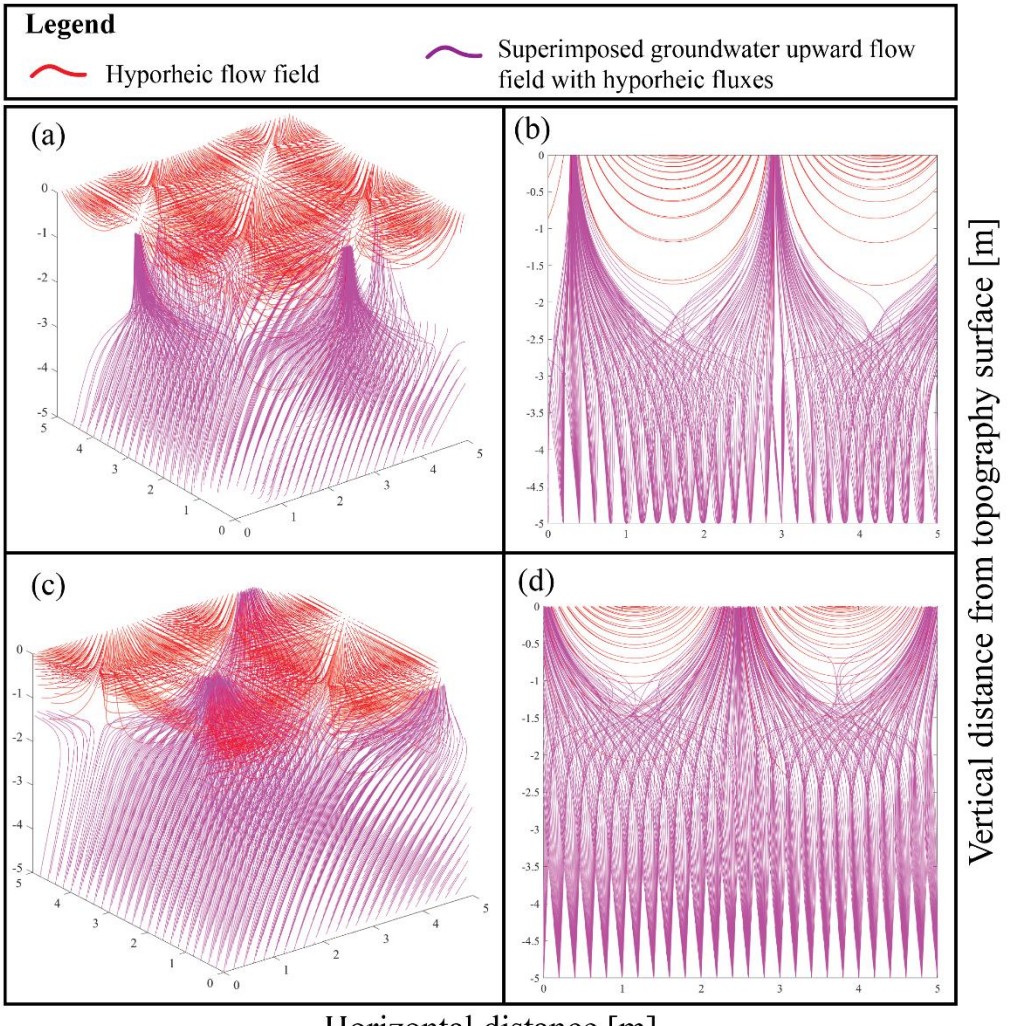

**Figure 8.** The impact of hyporheic fluxes (red lines) on the upwards groundwater flow (magenta lines) in the superimposed models (including both intermediate and deep groundwater flows). The figure corresponds to 2 arbitrary superimposed cases of a dynamic head coefficient of 0.08, **(a)** and **(b)** a hydrostatic damping factor of 0.73; **(c)** and **(d)** a hydrostatic damping factor of 0.13; **(a)** and **(c)** three-dimensional views; and **(b)** and **(d)** side views.

The distribution of the catchment-scale groundwater upwelling areas at the streambed interface was significantly affected by the streambed-induced fluxes. The fragmentation of groundwater upwelling is defined as a shift in the distribution of coherent areas of upwelling groundwater at the streambed interface towards smaller areas. To assess the fragmentation of groundwater





flow at the streambed interface, the distribution of coherent upwelling areas was quantified at the top surfaces of the streambed-scale model and catchment-scale model (i.e., without consideration of hyporheic flows) with resolutions of 0.1 m and 5 m, respectively. The fragmentation of the upwelling areas was presented as cumulative distribution functions, CDFs; a higher fragmentation shifted the corresponding CDF plot towards smaller areas, and it should be noted that the results do not consider the penetration depth of the groundwater flow path. The fragmentation results showed that in the absence of hyporheic flow

fields, 96% of the coherent upwelling areas of the regional groundwater flow were larger than the size of the streambed-scale model's top surface (i.e., 25 m$^2$) (Figure 9a, solid blue line) and that the median value at the top surface was 400 m$^2$ (Figure 9b, dashed blue line). On the other hand, when considering only the streambed-scale model (i.e., only incorporating hyporheic flow paths), the areas with coherent upwelling were found to be less than 7.5 m$^2$ (30% of the total streambed area) (solid red line in Figure 9). As expected, the upwelling of the streambed-induced flow was more fragmented than the upwelling of the

regional groundwater flow (shifting towards smaller areas in the CDF plots). The deep groundwater upwelling flow accelerated and thus contracted towards the topography surface, showing the variation in the upwelling flux area that occurs along the deep groundwater flow trajectories towards the topography surface. In particular, the ratio of the deep groundwater upwelling area at the release point ($Area_{(z=-500\,m)}$) to the deep groundwater upwelling area at the bottom of the sediment layer ($Area_{(z=-5\,m)}$) indicated the contraction of the flow area ranging between $1\times10^{-4}$ and $1\times10^{-2}$ (Figure S6 in the supporting

information). When taking the streambed-induced fluxes into account, the coherent areas of upwelling groundwater were substantially decreased, with approximately 95% of coherent upwelling areas in the superimposed models being less than 1.25 m$^2$ (5% of the top surface of the streambed-scale model) (Figure 9, solid magenta line). Thus, by comparing the fragmentation of groundwater upwelling areas with and without streambed-induced fluxes, the strong influence of hyporheic flows on the fragmentation of groundwater upwelling areas was revealed.

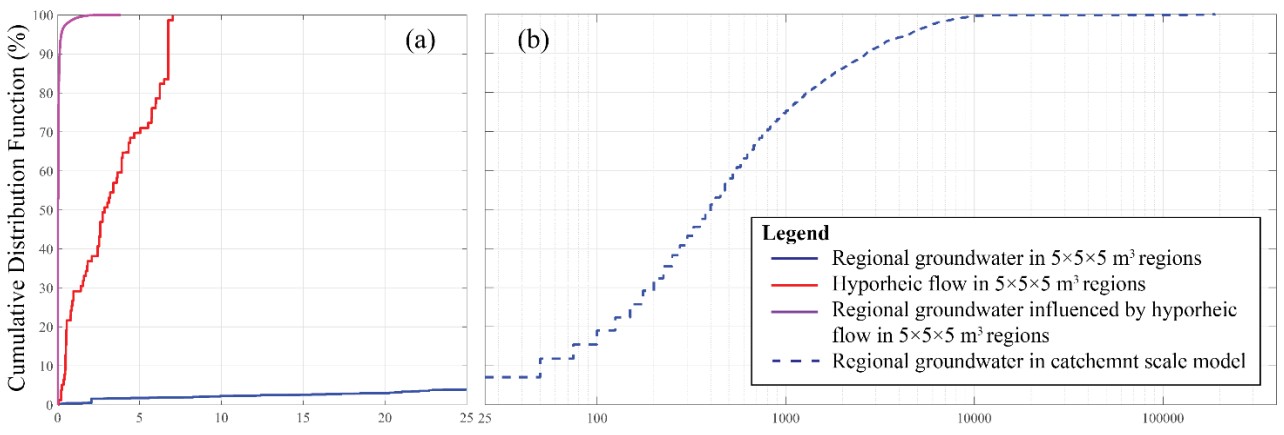


**Figure 9.** Cumulative distribution function plot of the coherent upwelling areas at the streambed interface for **(a)** groundwater flows (i.e., solid blue line), hyporheic exchange flows (i.e., solid red line), and groundwater flows influenced by hyporheic exchange fluxes (i.e., solid magenta line) in 5×5×5-m$^3$ regions and **(b)** groundwater flows (dashed blue line) in the catchment-scale model.





## 4 Discussion

Radionuclide safety assessments concern the possible leakage of radioactivity from a geological disposal site that is transported and subsequently discharged with deep groundwater into surface water where humans can be exposed to radiological doses through different uptake scenarios. The higher retardation of radionuclide compounds in Quaternary deposits and aquatic sediment compared to in bedrock fractures leads to prolonged radionuclide transport times in shallow subsurface regions (Marklund et al., 2008). One scenario implies that radioactivity accumulates in shallow aquatic sediments, where it may be
incorporated in agricultural products after future land use changes or by groundwater withdrawal used for irrigation purposes (Avila et al., 2013). In addition, the existence of radionuclides in streambed sediments can endanger aquatic habitats (Torudd and Saetre, 2013). Therefore, despite the shorter travel times of water in streambed sediments caused by shorter flow paths and higher velocities compared to the travel times in Quaternary deposits and bedrock layers, the large retardation of radionuclides in streambed sediments leads to prolonged radionuclide transport time in aquatic sediments, highlighting the
need to investigate the impacts of hyporheic exchange fluxes on deep groundwater flows in sediment layers. Specifically, the rate coefficients for the regolith stratum used in dose assessment models are generally defined by the ratio of the discharge velocity and regolith thickness (Xu et al., 2007), which is a variable that we have found to be highly increased by the presence of hyporheic flows. This means that the turnover of radioactivity in aquatic sediments may in fact be faster than previously calculated and that the total radioactive compound transport time could potentially be lower due to hyporheic flows compared
to the results of previous assessments (i.e., those that neglected the presence of hyporheic flow fields). Moreover, hyporheic flows spatially focus accumulated radioactivity to small areas that potentially lead to higher radiologic exposure. Hence, evaluating the impact of hyporheic exchange fluxes (as the main existing flow in streambed sediments) on regional groundwater flow fields is essential for understanding the ecology and biology of a streambed as well as the fate and transport of contamination emanating from the bedrock.

Generally, ambient groundwater affects the hyporheic depth, while the streambed-induced flow changes the upwelling groundwater in terms of both the flow velocity and flow direction near the bed surface (Cardenas and Wilson, 2006; Boano et al., 2009; Mojarrad et al., 2019a). The implication is that the exchange of natural solute elements and heat emanating from deeper groundwater will be affected by the presence of hyporheic flows; this is especially manifested in terms of very narrow pinhole flows from deep groundwater to surface water.

### 4.1 Regional Groundwater Flow Travel Time


The results of the catchment-scale modeling showed that the resolution of the topography of the groundwater table controls the modeled groundwater fluxes, similar to what has previously been found (Marklund and Wörman, 2011). The natural boundary condition consists of both topography-controlled head (at lakes and streams) and recharge-controlled head (at topography surfaces with downward flow directions) boundary conditions. The results highlighted the role of the DEM
resolution in infiltration zones to apply a head boundary condition that also satisfies the natural infiltration condition. In this





study, the groundwater discharge zones were found to align well with the river network (stands for 90% discharge points) and/or lakes (stands for 10% discharge points), which were topographically low points in the landscape with frequent relatively deep Quaternary deposits similar to those found by Marklund et al. (2008) and Caruso et al. (2016). In particular, deep groundwater flows mostly reached the topography surface downstream in the lowest part of the catchment, which also had a

lower slope and contained soils with higher hydraulic conductivity than the upper part of the catchment (see, e.g., Figure 4). The results of this study showed that the travel times of groundwater flows in bedrock with thicknesses greater than 450 m were in the range of 95-40,000 years, which confirms the finding of Ericsson et al. (2006). However, the median value of the groundwater travel time in the bedrock layer in our study slightly differs from what Ericsson et al. (2006) found (700 years and 1200 years, respectively). The difference between the median groundwater travel time in the bedrock domain determined

in this study and that determined by Ericssons et al. (2006) is due to the variation in the studied areas, which is reflected, e.g., in the thicknesses of the bedrock layers, as well as in the differences in the applied function describing how the hydraulic conductivity decays with depth. In addition, the travel times of the catchment-scale groundwater varies among different layers due to variations in the layer thickness, porosity, and hydraulic conductivity values. Furthermore, the radionuclide transport time is larger than the water travel time due to the retardation induced by sorption and diffusion processes. Radionuclide

retardation varies between different subsurface strata and depends on several parameters, such as soil porosity, the fracture size of the bedrock, and the ratio of adsorbed mass to dissolve mass within the subsurface layer (Kuilek and Nielsen, 1994; Xu and Wörman, 1999; Wörman et al., 2004). Hence, a higher retardation factor in Quaternary deposits and sediments compared to that in bedrock leads to the accumulation of radionuclides within aquatic sediment for long periods of time and increases the exposure time of radiological doses to humans. The retardation coefficient is usually considered a constant value for a

given material, but the spatial heterogeneity in the permeability of a single subsurface layer (e.g., sediment) leads to heterogeneous retardation coefficients (Piqué et al., 2013). In addition, the retardation coefficient is determined using breakthrough curves, BTCs (Li et al., 2009), in which increased groundwater flow velocities near the bed surface that occur due to the impact of the hyporheic flow fields change the shapes of the BTCs and, consequently, the values of the retardation coefficients of streambed sediments. Nested groundwater flow systems include subsurface flows across various spatial scales

with different flow properties (flow trajectories, flow velocities, etc.). The results showed a substantial difference in the sizes of the discharge areas of the regional groundwater and hyporheic flows at the streambed interface (Figure 9), in which the groundwater flow discharge area is significantly fragmented into smaller areas under the influence of hyporheic fluxes. Hence, in addition to the previously determined characteristics such as groundwater trajectories (Vissers and van der Perk, 2008; Wang et al., 2014) and the groundwater residence time distribution (Wang et al., 2016), the sizes of the discharge areas of

subsurface flows at the streambed interface can be used to distinguish the regional and hyporheic flows in hierarchically nested groundwater flow systems. The investigation of hierarchical nested groundwater flow systems improves our understanding of quantitative and qualitative groundwater flow-related phenomena, including the fate and transport of solutes and contaminants (Zijl, 1999).



### 4.2 Representativity of Streambed-Induced Flow Field

The streambed-induced flow field depends on the contributions of the hydrostatic and dynamic hydraulic heads to the head boundary condition at the streambed-subsurface interface. The stream water Froude number and streambed topography variation are the primary factors that control the hydrostatic and dynamic contributions to the pressure along the streambed (Tonina and Buffington, 2007; Käser et al., 2013). Mojarrad et al. (2019b) showed that the uncertainty in the hydrostatic damping factor has a substantial impact on the streambed-induced flow field. Field experiments reported in the literature

indicate that there is high uncertainty in the hydrostatic damping factor at spatial scales less than 20 m (Morén et al., 2017; Mojarrad et al., 2019b). In this study, a Monte Carlo simulation with 400 randomized combinations of hydrostatic damping factors (from a uniform distribution ranging between 0 and 1) and streambed topographies (from 20 different realizations) was used in a sensitivity analysis of hyporheic flows. In addition, the streambed-induced dynamic head was evaluated using the available discharge stations and the corresponding drainage area of each of 30 subcatchments (i.e., discharge clusters), along

with the mean values of the physical characteristics of the stream segments. Thus, the statistical (aleatoric) uncertainty in the hydrostatic and dynamic head contributions to the total hyporheic hydraulic head was covered using the applied Monte Carlo sampling approach. The fact that the hydrostatic head component dominated most of the streambed-scale total hydraulic head was consistent with the previous findings of Mojarrad et al. (2019b). In addition, the depth-decaying hydraulic conductivity of streambed sediments followed an exponential decay function according to the field investigation of Morén et al. (2017), which

was based on sets of hydraulic conductivity measurements at 3 cm and 5 cm within streambed sediments. The epistemic (systematic) uncertainty included in the hyporheic flow fields induced by the depth-decaying hydraulic conductivity function and the spatial heterogeneity of streambed sediments was not recognized in this study.

### 4.3 Dominating Behavior of the Streambed-Induced Fluxes

In hierarchically structured groundwater flows, upwelling groundwater is known to converge and accelerate towards discharge

zones, but our findings emphasize that this convergence phenomenon is particularly strong near streambeds due to the impact of the hyporheic fluxes (Figure 8). As a consequence, the distribution of the coherent groundwater upwelling areas was found to be fragmented into much smaller regions that resembled pinholes at the streambed where the deep groundwater flow penetrated through the hyporheic flow (Figures 8, 9). The results of the pinhole groundwater discharge pattern found in this study confirm the finding of Lidman et al. (2013), who observed small areas of groundwater discharge through a field

investigation using uranium, thorium, and radium measurements in the Krycklan catchment. Since the distribution of flow paths in the sediments is highly important for understanding the hydrological, biological and ecological processes at the streambed interface (Mathers et al., 2014; Mendoza-Lera and Datry, 2017; Fox et al., 2018), the revealed convergence of groundwater flow paths near the streambed interface is hypothesized to be of high importance for groundwater-surface water exchange processes. The temperature differences between deep groundwater flows and hyporheic exchange fluxes (Mamer

and Lowry, 2013) reflect a pronounced spatial focusing of the discharge zones of heat and mass transfers emanating from deep





groundwater due to the strong pinhole behavior of groundwater at the streambed interface. The degree of fragmentation substantially depends on the magnitudes of the streambed-induced fluxes and the vertical velocities of groundwater flows. Therefore, it is clear that this fragmentation of groundwater discharge is much more pronounced in stream water than in lake bottoms that are less subjected to hyporheic flows. In addition to the hyporheic and regional groundwater flow velocities, the

heterogeneity of a subsurface and, in particular, how the hydraulic conductivity decays with depth have primary controls on the distribution of hyporheic flow paths (Earon et al., 2020) and thus also strongly influence the depth where the contraction of the near-bed groundwater flow paths begins. Furthermore, due to flow continuity, the velocity of a flow increases if the area decreases. Thus, as the groundwater flow converges into smaller areas due to the impact of streambed-induced fluxes, the flow velocities increase for both intermediate and deep groundwater. The travel time of the flow has a reverse relation with the flow

velocity, meaning that higher velocities decrease the flow travel times of aquatic sediments (Figure 7). An implication of these results is that relatively shorter groundwater flow travel times within the streambed sediment impact the hyporheic habitat diversity and, consequently, stream water quality (Poole et al., 2006). A particular effect is due to the transport of radionuclide leakage from deep geological waste disposal sites; the accumulation of radionuclides in stream water sediment depends on (a) the location and size of the groundwater upwelling discharge areas and (b) the deep groundwater travel times in the streambed

sediments. Our results indicated that streambed-induced fluxes slightly relocate the groundwater upwelling discharge points (Figure 8). This is due to local spatial scales connected to streambed-induced flows. However, more importantly, the hyporheic flow fragments the groundwater upwelling areas and induces shorter groundwater travel times near the bed surface. The convergence of deep groundwater discharge in the landscape demonstrates a reduction in the area contaminated by radionuclides that may possibly leak from an HLRW in the future. The radionuclide rate transfer coefficient in the compartment

model is inversely proportional to the groundwater travel time and upwelling discharge area (Xu et al., 2008; Wörman el al., 2019); the shorter travel time near the bed surface and smaller size of the coherent upwelling areas result in a higher transfer rate coefficient (Figure S7, supporting information).

## 5 Conclusion

This study quantified the impacts of hyporheic fluxes on groundwater discharge in surface waters, with specific attention paid

to the influence on deep groundwater upwelling flows that carry radionuclide compounds from waste repositories placed in deep bedrock. The main findings of the paper are as follows.

- Hyporheic fluxes substantially influenced groundwater travel times close to streambed surfaces. Specifically, the results indicate that both intermediate (not circulating in bedrock) and deep (circulating in bedrock) groundwater travel times decreased by 50% through the top 5 m of aquatic sediments compared to when the hyporheic fluxes were

530       omitted.

- The groundwater upwelling areas strongly contracted near the streambed surface due to the impact of hyporheic fluxes, thus appearing as narrow "pinholes" of groundwater discharge. This phenomenon of the strong convergence





of groundwater flows in stream bottoms led to a distribution of very small and fragmented discharge areas. Due to
flux continuity, the groundwater flow velocity increased when the upwelling areas shrank, resulting in a maximum
groundwater vertical velocity at the streambed interface.

- The contraction of the groundwater discharge at the streambed interface is controlled by the relative intensities of
  groundwater circulation and hyporheic flows, the heterogeneity of the streambed sediments, and the depth-decaying
  hydraulic conductivity of the streambed sediments. This phenomenon is important for the exchange of solute elements
  and heat between deep groundwater and streams and hence for general stream ecology, such as the formation of
suitable benthos habitats.

- Despite the large differences observed in the depths and deep groundwater flow velocities among the three different
  subsurface layers (i.e., bedrock, Quaternary deposits, and sediment), the radionuclide transport time distribution is
  relatively similar among the layers due to the retardation impact induced by sorption and diffusion processes.

- The groundwater travel time reduction and the contraction of the groundwater flow near the bed of the surface would
lead to a higher radionuclide transfer rate coefficients for discharge zones in biosphere compartment models.
  However, the higher spatial focusing of upwelling is an additional important aspect for radiological doses transmitted
  to humans and other living species.

- The uncertainty in representing dynamic and static head components driving hyporheic flows was determined using
  a Monte Carlo simulation with stochastic distributions of the streambed topography and the hydrostatic damping
factor. The results demonstrated a small control on the hyporheic flow of the dynamic head component in relation to
  the hydrostatic one due to the marked subcritical flow condition in the considered stream network. The results also
  showed that the relative importance of the streambed dynamic head component increased with the Froude number,
  which could vary substantially over time with hydrologic conditions as well as among streams.

- The deep groundwater discharge points closely follow the hydrological drainage system in a landscape. The deep
groundwater flow paths are subjected to large-scale convergence towards discharge zones and predominantly end up
  in a few areas in the downstream region of the studied catchment. Downstream regions of the catchment often coincide
  with regions of relatively deeper Quaternary deposit layers. This reflects the role of topography and the characteristics
  of Quaternary deposits in regional groundwater circulation, which may further highlight the significance of
  Quaternary deposits in this context.


*Author contributions.* Conceptualization, B.B.M., A.W., and J.R.; Investigation, B.B.M.; Formal analysis, B.B.M.; Funding
acquisition, A.W.; Visualization, B.B.M.; Supervision, A.W., J.R.; Writing − original draft preparation, B.B.M.; Writing −
review and editing, B.B.M., A.W., J.R., Sh.X.

*Competing interests.* The authors declare that they have no conflicts of interest.





*Acknowledgments.* This study was financed by the Swedish Radiation Safety Authority, SSM, and EU's Horizon 2020 Research
and Innovation Programme under Marie-Skłodowska-Curie grant agreement No. 641939. All the data can be accessed from
the cited references or are made available at the Krycklan website (https://www.slu.se/krycklan); no new data were created to
generate the results of this manuscript.

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
