# Peer review of "The influence of hyporheic fluxes on regional groundwater discharge zones"

_Hydrology and Earth System Sciences, 2021_

## Author Comment (AC1)

**Response to Community Comments #1 by: Ulrik Kautsky**

- The paper presents an interesting model study concerning potential effects of the hyporheic zone on groundwater flow patterns using a selection of the hydrological data available for Krycklan. In its present state the study is theoretical. The study would be strengthened if it included an analysis describing the extent to which the regional model was able to capture the hydrology at Krycklan. This would help to address questions about the validity of the regional hydrological model and may help frame results from the hyporheic, local-scale simulations as applicable to a real-world scenario.

  The paper would be further strengthened if it discussed the robustness of the results as affected by the parameterizations and structural assumptions made in the numerical and conceptual models. For example, the potential effects of landscape topography, time variation of streambed topography, rock fractures, soil stratification, spatial heterogeneity of streambed sediments, parameterization of the infiltration rate at the surface, and model boundary conditions are not discussed. Without such a discussion it cannot be deduced the extent to which variabilities in these, and other, variables will affect the hyporheic phenomena postulated in the modelling exercise; it is therefore not possible to assess the extent to which the results and/or methodologies presented in the study are relevant to other sites.

Thanks for the comments. This study was conducted in a specific watershed as a site-specific limitation, in terms of landscape topography, the geological heterogeneity of soil strata, and the available infiltration. Even though a sensitivity analysis using more parameters than were used in the presented Monte Carlo analysis could have been useful, the purpose was not to derive general conclusions that can be unconditionally used in other areas. In addition, the present study recognized uncertainties in the ways in which hydrostatic and dynamic head boundary conditions were represented in the model. The influences on the modeled results are analyzed using Monte Carlo simulations, which cover the uncertainties of the spatial and temporal variations in streambed properties with 1200 realizations. The uncertainty in this study is addressed in section 2.5 (lines 217-222). The Krycklan catchment can be divided into a number of sub-catchments. In this study, the modeled water travel times for the intermediate groundwater flowing from recharge areas to discharge points throughout the whole catchment are presented in Fig. 7. The median value is approximately 2.5 years, which is in the same order of magnitude as the results (1.2–7.7 years) obtained by a recently published study performed at the same catchment area (Jutebring Sterte et al., 2021). Jutebring Sterte et al. (2021) used a combination of observations of $\delta^{18}O$ and base cation concentrations in different streams, then conducted particle tracing (flow modeling using MIKE SHE II) to provide a consistent picture of the hydrological functioning of the Krycklan catchment. The following text will be added to the revised version (in section 4.1):

> *In this study, the modeled water travel times for the intermediate groundwater flowing from recharge areas to discharge points throughout the whole catchment are presented in Fig. 7. The median value is approximately 2.5 years, which is in the same order of magnitude as the results (1.2 − 7.7 years) obtained by a recently published study performed in the same catchment area (Jutebring Sterte et al., 2021).*

In addition, previous studies have shown the deterministic effects of topographical variation compared to heterogeneity in the fracture network within the bedrock domain on deep groundwater travel times in different subsurface strata, as well as their discharge zones at the topographical surface (Selroos et al., 2002; Marklund et al., 2008; Welch et al., 2012). Therefore, the uncertainty involved in the fracture network has a negligible effect on the results of this study. The following text will be added to the revised paper in order to address this comment:

> *Previous studies indicated that the deep groundwater discharge location at the topographical surface is primarily controlled by topographical variation, and the distribution of the fracture network may slightly change the discharge location (Selroos et al., 2002; Marklund et al., 2008; Welch et al., 2012). In this study, the bedrock domain was assumed to be an equivalent continuum subsurface stratum, where an average hydraulic conductivity with an exponential depth-decaying function (Figure 1) was considered to represent both the intact bedrock and fractures.*

**References**

Jutebring Sterte, E., Lidman, F., Lindborg, E., Sjöberg, Y., and Laudon, H.: How catchment characteristics influence hydrological pathways and travel times in a boreal landscape, Hydrology and Earth System Sciences, 25, 2133-2158, 2021.

Marklund, L., Wörman, A., Geier, J., Simic, E., and Dverstorp, B.: Impact of landscape topography and Quaternary overburden on the performance of a geological repository of nuclear waste, Nuclear technology, 163, 165-179, 2008.

Selroos, J.-O., Walker, D. D., Ström, A., Gylling, B., and Follin, S.: Comparison of alternative modelling approaches for groundwater flow in fractured rock, Journal of Hydrology, 257, 174-188, 2002.

Welch, L. A., Allen, D. M., and Van Meerveld, H.: Topographic controls on deep groundwater contributions to mountain headwater streams and sensitivity to available recharge, Canadian Water Resources Journal/Revue canadienne des ressources hydriques, 37, 349-371, 2012.

---

## Author Comment (AC2)

**Response to Comments from Referee #2**

- The paper "The influence of hyporheic fluxes on regional groundwater discharge zones" submitted to HESS by Mojarrad et al. investigates different scale flow systems by means of topographically induced groundwater circulation and the role of hyporheic flow on groundwater circulation in the river valleys. The flow systems are discussed in relation to possible accumulation of radioactive waste originating from deeper aquifer regions within the Quaternary deposits; by differentiating between re- and discharge zones.

We thank the referee for providing a thoughtful review of our paper. Our detailed reply to the comments (both major and minor) is provided below. The paper will be revised after the online discussion, and changes are presented in *underline text in italic.*

- I think that the approach, of the influence of topography on the distribution of hydraulic potential in different scales is interesting. The extent to which the link of deep groundwater circulation with the potential field of the hyporheic interstitial can be easily connected would have to be discussed in more depth, especially by delving into the processes that can also influence the potential fields at different scales.
  The concept underlying the work is not new and it is surprising that the authors do not mention the fundamental work of J. Toth.

Thanks for the comment. We will include a new section at the beginning of section 2.6 of the revised paper to describe the extent of the linkage between deep groundwater and hyporheic flow, as follows:

> *Particle tracking was used to identify streamlines that follow hyporheic, intermediate and deep groundwater flows. Deep groundwater was defined as flow that entered the bedrock, and intermediate groundwater was determined to only be present in Quaternary deposits. Hyporheic flow contains streamlines starting and ending in the stream bottom within a 5 m spatial scale. Particle tracking was conducted at two different spatial scales: the regional scale (i.e., entire catchment) and the local scale (i.e., $5 \times 5 \times 5$ m³). The regional-scale particle tracking was conducted to evaluate deep groundwater discharge zones, as well as to distinguish the deep and intermediate groundwater flow fields (see section 2.6.1). In addition, particle tracking was conducted on a large number of $5 \times 5 \times 5$ m³ cubes containing a hyporheic flow field in deep groundwater discharge zones, in the absence and presence of upwelling groundwater (both deep and intermediate flows), in order to investigate the impact of hyporheic flow on deep groundwater discharge.*

We completely agree that full credit should be given to the previous research conducted by Tóth. However, the main differences between our study and previous research (including Tóth (1963)) should also be better addressed. Therefore, a new section will be added in the Discussion (i.e., section 4.1) to address the hierarchal structure of the subsurface flow systems, as below:

> *Nested groundwater flow systems comprise subsurface flows across a wide range of spatial scales, all with different flow properties. The applied multiscale approach of this study addresses regional, intermediate and hyporheic flow systems, which is consistent with the hierarchically nested groundwater flow cells first described by Tóth (1963). In addition to basin geometry and the surface topography, as described by Tóth (1963), the geological heterogeneity of the subsurface hydraulic conductivity (as dependent on stratification and soil type) significantly impacts the local, intermediate and regional groundwater flow discharge zones (Jiang et al., 2010, 2011). Further, this study also separates the treatment of discharge and recharge zones by applying different top boundary conditions, and the local-scale investigation focuses on the boundary conditions that arise in surface water systems and impact hyporheic flow. These interactions are not considered in the previous literature incorporating the subsurface flow nested perspective.*

- The authors cite the work of W. Zijl, who, through a Fourier analysis and consideration of the anisotropy of the geologic sequences, or important boundaries, such as the topography of the bedrock surface, or different structural properties of the bedrock, determine a series of flow cells. A reduction to two or three as proposed in this paper is probably too simplified. Thus, I would find it justified for the authors to emphasize the conceptual aspect of the work and to discuss the role of influence of topography of the river bed (i.e., the river corridor concept of Stanford and Ward), the influence of heterogeneity of the geological sequence, the role of the relative water flux contributions etc.).
  Based on the work (concept) of Toth, it is not surprising that the drainage system can correspond to the exfiltration zones.

However, there are other aspects to be considered:
The River corridor concept of Stanford and Ward, shows that the topography of the riverbed and the topography of an impounding layer (can be e.g. glacial deposits, bedrock surface or discontinuities of the gradient of the riverbed along a river course) produce infiltration and exfiltration zones within the hyporheic interstitial, which can be very dominant with respect to water fluxes.
I do not know the glacial sediments of the area, but I suspect that because of the diversity of processes, heterogeneities in these deposits also lead to vertical hydraulic gradients that could significantly affect the simple potential distribution.

This study divided the flow analysis into two scale ranges, the "regional-scale" and the "streambed induced flow"; however, the representation of the top boundary condition included a range of scales covering these two arbitrarily (but justifiably) separated aspects of the flow problem. Since this study considers a wide range of scales for the head boundary condition, we think that it describes a similar flow phenomenon to the river corridor concept of Stanford and Ward (1993). The following section will be added in section 4.1 to address the referee's comment:

*Only decreasing the DEM resolution over the recharge zones, while maintaining the high DEM resolution (i.e., 2 m × 2 m) in discharge zones (i.e., stream network, lakes, etc.), allows us to evaluate in- and exfiltration zones via discontinuities in topographical gradient throughout the river network in the catchment-scale model (Stanford and Ward, 1993).*

- The structural heterogeneity of the bedrock and the character of the hydraulically relevant structures of the subsurface (shear zones, fracture patterns, etc.) can also significantly affect the anisotropy of the hydraulic conductivities, so that over long geologic times the pattern of the exfiltration zones is also not necessarily uniformly distributed along the drainage system.
- The concept depth decaying hydraulic conductivity has to be approved by regional specific data. Other, more complex heterogeneity development with depth could have a strong influence on the Potential distribution.
- A much more important point influencing the regional flow systems over longer times periods (transport of radionuclides from certain depths of the bedrock) is the dynamics of the development of the topography. Please discuss, how topography was shaped during the last 15'000 years. Over long time periods, topography cannot be assumed to constant.
- **I think the paper could benefit from a more in-depth discussion. In its current form, the argumentation is a bit too simplistic.**

Thanks for the comment. The heterogeneity of the bedrock domain was determined by applying depth decaying hydraulic conductivity functions. In addition, previous studies showed the controlling effects of topography variation compared to heterogeneity in the fracture network within the bedrock domain, as regards the travel time of deep groundwater in different subsurface strata as well as their discharge zones at the topographical surface (Selroos et al., 2002; Marklund et al., 2008; Welch et al., 2012). The following discussion is added to the revised paper to address the referee's comment:

*Previous studies have indicated that the deep groundwater discharge location at the topographical surface is primarily controlled by topographical variation, and the distribution of the fracture network only slightly affects the discharge locations (Selroos et al., 2002; Marklund et al., 2008; Welch et al., 2012). In the absence of detailed data for the study region, the bedrock was assumed to be a continuum subsurface stratum with average hydraulic conductivity, and the horizontally constant hydraulic conductivity was assumed to decay with depth (Figure 1). In the present study, the applied depth-decaying hydraulic conductivity function was based on regional hydraulic conductivity field measurements conducted by Swedish Nuclear Fuel and Waste Management, SKB (Ericsson et al., 2006). Horizontal variability in the hydraulic conductivity was assumed, but only for the overlying soil strata (Quaternary deposits and possible aquatic sediments). These uppermost layers have a dominating effect on the discharge of groundwater, and thus, we believe that this simplified model represents the most important patterns in hydraulic conductivity.*

We agree with the referee that the topography is not constant over a long time period, and that this needs to be discussed as regards groundwater discharge as well as solute/contamination transport over long distances. This region is primarily affected by retreating glaciation, which has led to initial glacial processes followed by sea sedimentation and substantial landrise, with a change in ambient groundwater flow. Therefore, we expanded the discussion of the implications of the dynamic behavior of the topographical and subsurface geological properties over a long time period in our results. However, for the assessment of the flow in discharge zones, which is the main focus of the paper, non-stationarity is of less importance, and a quasi-steady condition might be more

relevant. This is because of the quick adaptation of the groundwater flow to specific changes in the boundary condition—non-stationarity is primarily reflected in the continuity equation and the associated change in groundwater storage over long time. The following will be added into section 4.1 to address the referee's comment:

*In this study, the bedrock domain was assumed to be an equivalent continuum subsurface stratum, where average hydraulic conductivity with an exponential decay function (Figure 1) was assumed to represent both the intact bedrock and fractures. Depth-decaying hydraulic conductivity has previously been suggested for the Krycklan catchment (e.g., Ameli et al., 2016b). Moreover, in the present study, the applied depth-decaying hydraulic conductivity function was based on regional hydraulic conductivity field measurements conducted by Swedish Nuclear Fuel and Waste Management, SKB (Ericsson et al., 2006). Topography and bedrock evolution occurred from 3000 Mega-annum ago (Mega-annum abbreviated as Ma, where 1 Ma is equal to 1 million years) (formation of basement rock) to less than 1 Ma ago (glacial erosion, slope formation, as well as frost process) (Lidmar-Bergström, 1996); in this time, glacial erosion could have changed the bedrock formation and surface topography by up to 600 m in northern Sweden (Lidmar-Bergström, 1997). About 10,000 years ago, retreating glaciation initially led to a glacial process followed by substantial landrise, with successive changes in regional groundwater flow. These factors may hold significance as regards the transport along regional flow paths. However, the locations of discharge zones at the topographical surface adapt relatively quickly to changes in boundary conditions, and the location of discharge follows the valleys in the low lands, which could slightly vary within less than 3 Ma in northern Sweden (Lidmar-Bergström, 1996). Since our results for groundwater transport time (Figure 5) are in the range of hundreds-thousands years, they are representative of stationary/quasi-stationary conditions, but not necessarily of a specific historic or future period's transport scenario.*

**General Comments**

- Although the character of the model is mostly conceptual the authors state that results from intense field investigations exist. Nevertheless, nearly none of these existing data are specified or used for calibration and/or validation of the model (i.e. character of the hyporheic zone, heterogeneity character of the glacial deposits.

The existing data for independent parameters were used in the development of the numerical model, such as hydraulic conductivities, channel properties, topography and bedrock surfaces, etc. However, very few data on the dependent flow field exist. The interaction between deep groundwater flow and hyporheic fluxes was investigated using the independently developed model over a wide range of spatial scales of regional groundwater flow. However, the limited data necessitated the use of Monte Carlo simulation as a sensitivity analysis to cover the uncertainty of the hyporheic flow evaluation. Hence, the study's aim was to provide a more comprehensive picture of the investigated phenomenon, rather than being a case-specific investigation.

- For seven different soil types in the Krycklan catchment hydraulic conductivity was obtained. A sensitivity analysis for hydraulic conductivity of streambed sediment would be interesting.

Thanks for the comment. Previous studies have reported streambed sediment hydraulic conductivity in the range of $10^{-3}$-$10^{-5}$ (m/s) (Wörman et al., 2002; Salehin et al., 2004; Ryan and Boufadel, 2006; Song et al., 2007), regardless of the dominant soil type of the adjacent area. Therefore, in this study we used the results of a field investigation conducted by Morén et al. (2017) on a 1500-meter stretch for estimating the hydraulic conductivity of the streambed sediment. However, it would be great if a sensitivity analysis could be performed using other field experimental results in the future.

- Provide a more quantitative of visualization of discharge locations (Fig. 1), e.g. by means of point densities. Likewise, an illustration with the "pinholes" of groundwater discharge and/or "nested" flow cells (maybe for a zoom) would also help to understand the different flow processes.

Figure 1 shows only the hydraulic conductivity and the porosity of the different subsurface domains. If the referee meant to refer to Figure 2 (which shows the nested flow system), this figure does not present results of the analysis, but instead tries to illustrate the applied method. However, the paper will be revised after the online discussion, and Figure 1 and Figure 2 will be combined. It should also be mentioned that the result concerning nested flow cells (containing groundwater and hyporheic flow fields) within the streambed sediment had already been illustrated in Figure 8. In addition, Figure 7 shows the quantitative results of the impact of hyporheic fluxes on deep and intermediate upwelling groundwater at the deep groundwater discharge points in a nest flow system within the streambed sediment.

- How was the catchment area delineated (surface?). And is this approach appropriate when evaluating the deep aquifer, when the shape of the topography changed over the last 15'000 years? A 3D visualization would help to better understand the geological settings in relation to the topography.

The catchment area was derived from a basin that had already been delineated in previous research (Laudon et al., 2013), which used the topography DEM file to identify the upstream area that contributes to a common outlet draining the region. The following text will be added to describe the extension of the model in the revised version of the paper:

> *The bounding rectangle of the Krycklan catchment (i.e., 11.6 × 10.3 km²) was set as the horizontal limit of the numerical model's domain (64°11.8´N -64°17.6´N, 19°39.5´E-19°54.3´E).*

Comparing the depth of the subsurface layers (the depth of Quaternary deposits was almost 40 m, and the deep groundwater was traced from 500 m depth) with the extent of the domain (11.6 × 10.3 km²) shows that the catchment area is relatively far from the lateral boundaries. In addition, in our study, a constant hydraulic head was assumed for the lateral surfaces of the catchment-scale model in order to incorporate the effect of the sides of the numerical model on groundwater flow circulation. Therefore, part of the deep groundwater flow (represented by the particles in the catchment-scale particle tracing) showing upward flow at 500 m left the domain via the side walls (particle tracing statistics already presented in Figure S4, supporting information). As was mentioned earlier in response to the referee's main comments, a new section of text will be added into the revised version of the paper, where the dynamic behavior of the topography will be discussed.

- Some repetitions could be avoided, like e.g. Software use.

Thanks for the comment. The comment of the referee will be addressed by removing the repetitions.

**Detailed Comments**

- L29: Definition of "long-term"

Thanks for the comment. The definition of "long-term" in this paper will be added in the revised version, as follows:

> *Long-term radionuclide safety assessment regards the maximum consequences of radionuclide compounds for human health and the environment over a time span up to 1 million years after the radionuclide waste repository has been closed (Kautsky et al., 2013).*

- L54: This strongly depends on the geology which means that without a regional geological model a detailed statement due to deep groundwater discharge zones remain fragmentary

Thanks for the comment. The sentence will be revised and supported by other studies, as follows:

> *Values founds in the literature indicate that the ratio of regional groundwater flow to hyporheic exchange flow in the streambed sediment is in the range of $10^{-2}$-$10^{-4}$ (Bhaskar et al., 2012; Goderniaux et al., 2013; Gomez-Velez et al., 2014). This suggests that local hyporheic flows can have significant effects on deep groundwater discharge zones, and thus on the solutes and heat transport associated with the flow of deep groundwater into streams.*

- L66: please add References of S. Todd

The referee did not provide information for the suggested references, and we could not find the appropriate references.

- L84: the history of the hydrology at different time scales will also influence the contribution of water from different flow systems in the exfiltration zones.

Thanks for the comment, which we agree with. An explanation will be added in the Discussion section of the revised version, as below:

*In this study, the topography-controlled boundary condition was considered at the discharge location in a stationary model. Therefore, the role of hydrological variation over different time scales, which could potentially affect the contribution of water from various spatial scales in subsurface discharge regions, was not investigated (Dam et al., 2012). However, due to the quick adaptability of groundwater flow to variations in boundary conditions, the non-stationary condition is less important to instantaneous groundwater discharge in the case of groundwater flow with a travel time in the range of hundreds to thousands of years.*

- L82: I cannot find information on geological heterogeneity

The geological heterogeneity is addressed in section 2.2 (i.e., empirical and observational data), where the seven different soil types are used to describe the spatial heterogeneity of Quaternary deposits. In addition, the depth-decaying hydraulic conductivities were considered in all the modeling domains (i.e., streambed sediment, QD, and bedrock).

- L86: Darcy's law should be known to the readership?

The definition of Darcy's law will be added to the paper, as follows:

*The momentum equation for subsurface flow in saturated porous media at a sufficiently low Reynolds number is generally derived by neglecting inertia terms, while maintaining the potential energy and adopting a linear friction-loss relationship. This leads to the well-known Darcy's law: $q = [U, V, W] = -K\nabla H$. Here, $K$ (m/s) is the hydraulic conductivity, $\nabla$ is the nabla operator, $H$ (m) is the total hydraulic head, $q$ (m/s) is the Darcy velocity vector, and $U$ (m/s), $V$ (m/s) and $W$ (m/s) are the Darcy's velocities in the x, y, and z directions, respectively, while bold symbols denote vector quantities (Whitaker, 1986).*

- L103: Missing specific information on the time scale and the rates of considered processes

We do not fully understand this comment. Line 103 of the paper refers to a description of the seepage velocity concept that was applied in the paper. We assume that "time scale and the rates of considered processes" refers to lines 105-107, where the retardation of the solutes due to adsorption and diffusion processes is described. In this section, the objective is to use the concept of the retardation of solute transport due to sorption to highlight the fact that the solute transport velocity is lower than the water seepage velocity, and, consequently, the transport time of the solutes in porous media is generally greater than that of seepage water through pores and fractures.

- L118: There should be drillings for geological information's. A corresponding map and a stratigraphic-lithologic overview, allowing to evaluate the degree of vertical variability of hydraulic properties is missing.

Unfortunately, we could not find quantitative geological information (i.e., borehole drilling information) for the Krycklan catchment. However, Sterte et al. (2018) and others provided an overview of the soil stratification in the Krycklan catchment. We will add the following to address the comment of the referee:

*The Krycklan landscape was formed during the last glaciation (Lidman et al., 2016); the northern part primarily consists of a 15-20 m thick glacial till. The glacial till intertwines with regions containing peat and/or lake sediment towards the east. The glacial till mainly contains basal till in the deep soil, which is replaced with ablation till in the shallow soil (Jutebring Strete et al., 2021). The southern side of the Krycklan catchment has a lower elevation compared to other regions, where the soil is a mixture of fluvial and glaciofluvial deposits. Those regions mainly consist of sandy and silty sediments with significant soil thickness (i.e., approximately 40 m deep), where the soil has been compacted by its own weight (Lyon et al., 2011). Generally, the aggregates of till soil shrink with depth, and in the sandy silty sediment, sand is replaced with silty clay as the depth increases. In addition, peat is replaced with clay within a few meters of the surface (Sterte et al., 2018).*

- L122: a sedimentological decription of the glacial sediments would allow a better info on heterogeneity, till compris a lot of different glacial sediment types

As was stated in a previous response, a brief description of the different glacial sediment types in the Krycklan catchment will be added to the revised version of the paper, as follows:

*The Krycklan landscape was formed during the last glaciation (Lidman et al., 2016); the northern part primarily consists of a 15-20 m thick glacial till. The glacial till intertwines with regions containing peat and/or lake sediment towards the east. The glacial till mainly contains basal till in the deep soil, which is replaced with ablation till in the shallow soil (Jutebring Strete et al., 2021). The southern side of the Krycklan catchment has a lower elevation compared to other regions, where the soil is a mixture of fluvial and glaciofluvial deposits. Those regions mainly consist of sandy and silty sediments with significant soil thickness (i.e., approximately 40 m deep), where the soil has been compacted by its own weight (Lyon et al., 2011). Generally, the aggregates of till soil shrink with depth, and in the sandy silty sediment, sand is replaced with silty clay as the depth increases. In addition, peat is replaced with clay within a few meters of the surface (Sterte et al., 2018).*

- L134: There is no information about the geometric extension of the model and why was Comsol chosen for such a simple model of the geological setting and not modflow, feflow or any groundwater affine software?

In general, other software could have been used, but based on our previous experience, we used COMSOL. Specifically, this software offers several key functionalities, such as the ability to apply different boundary conditions (such as the topography and recharge boundary conditions) within a single model, the ability to use mathematical functions as input for the hydraulic conductivity decay in different subsurface layers, improved nonrectangular mesh types, the ability to export results into MATLAB (for superimposing the hyporheic and regional-scale models), etc. In addition, COMSOL is a fully 3D software, whereas Modflow is only semi-3D. Hence, COMSOL was used as the numerical modeling software for our research, the focus of which is the upward flow in the discharge zones.

The geometric horizontal limits of the model will be added in the revised version of the paper, as below:

*The bounding rectangle of the Krycklan catchment (i.e., $11.6 \times 10.3$ $km^2$) was set as the horizontal limit of the numerical model domain (64°11.8´N -64°17.6´N, 19°39.5´E-19°54.3´E).*

- L137: The presented material does not allow this simplification, which project data support the statements of the cited publications?

The choice of depth-decaying hydraulic conductivity can be supported by the semi-analytical solution applied by Ameli et al., (2016b), which used the Krycklan catchment as its study domain. Moreover, although not specific to our study area, Ericsson et al. (2006) investigated the deep groundwater flow circulation in eastern småland, Sweden (using regional groundwater flow modeling). They measured the hydraulic conductivity of the bedrock domain at several depths, and fitted a depth-decaying exponential function representing the hydraulic conductivity within the bedrock. In addition, the observational data of hydraulic conductivity derived by Swedish Nuclear Fuel confirm the depth-decaying hydraulic conductivity in the bedrock domain (Swedish, N. F., 2008). Ingebritsen and Manning (1999) investigated the permeability of the continental crust using geothermal data, and identified a depth-decaying behavior in the permeability of the soil. We support our assumption regarding the depth-decaying hydraulic conductivity of the streambed sediment with a number of field investigation (e.g., Ryan and Boufadel, 2006; Song et al., 2007).

A new section of text will be added to the revised version of the paper to address the referee's comment, as follows:

*Previous studies have shown that hydraulic conductivity decays with depth for most geological features (Ingebritsen and Manning, 1999; Saar and Manga, 2004; Jiang et al., 2009; Grant et al., 2014; Ameli et al., 2016a; Ryan and Boufadel, 2006). In particular, Ameli et al. (2016b) used a semi-analytical approach to determine the depth-decaying hydraulic conductivity in the Kryclan catchment.*

- L158: The information of the mesh sizes should be expanded with the number and delaunay quality of elements

The details pertaining to the mesh size and the quality of the mesh elements will be updated in the revised version of the paper, as follows:

*A nonuniform mesh was used for the flow calculation of each layer. The mesh sizes varied within the ranges of 0.1-2 m, 2-17 m and 17-403 m for the streambed sediment, Quaternary deposit and bedrock layers, respectively. In addition, the maximum element growth rate and the curvature factor for the streambed sediment are 1.15 and 0.1, while these change to 1.2 and 0.2 for Quaternary deposits and 1.35 and 0.3 for streambed sediment, respectively.*

- L175: DT has not been introduced

$D_T$ is the total depth of the domain, as already defined: "the total depth of the domain, $D_T$, varied spatially within the catchment….".

- L178: give a reference to the particle tracing routine used

A reference to Genel et al. (2013) will be added to the revised paper.

- L181: One of the described aims in the introduction is the influence of radionuclide spreading for humans and biota. With a residence time of 320 million years it would be hard to make any predictions at all? A cumulative curve from particles reaching the surface over time could be more detailed.
  At the long time scale the topography has changed and as a result the flow field also would have changed.

The study was performed to investigate the effects of the hyporheic zone on groundwater discharge, and to show how it can affect the transport of radionuclides in shallow sediments below the surface of the water. The influence on humans and biota is beyond the scope of this study, but a potential connection was highlighted.

The particle tracing was conducted over a fictitious time period of 320 million years to allow all the particles to either leave the domain via the lateral sides or reach the top surface of the model. However, the maximum travel time of the groundwater flow was found to be approximately 40000 years (Figure 5). A cumulative curve reflecting the travel time of deep groundwater within each subsurface domain (from a depth of 500 m up to the topographical surface) has already been presented in Figure 5. Considering the range of travel times (100-40000 years in Figure 5), non-stationarity is of less importance, and a quasi-steady condition may be more relevant. This is because of the quick adaptability of the groundwater flow to a specific change in the boundary condition—non-stationarity is primarily reflected in the continuity equation and the associated change in groundwater storage over a long time. However, as was mentioned earlier in response to the comment of the referee, a separate discussion on the dynamic behavior of the geological formation will be added to the revised paper.

- L219-220: Specify "other models and statistical uncertainties"

Thanks for the comment. The sentence will be revised as below:

*This study recognized uncertainties in the hydrostatic and dynamic head boundary conditions by performing a sensitivity analysis on the parameters in Equation (5), while the uncertainty in the other parameters of the hyporheic flow model (such as hydraulic conductivity, etc.) and in those of the catchment-scale flow model was not formally analyzed.*

- L239: Correct "Monte Carla"

The word will be corrected in the revised paper.

- L335: Can you better explain the role of the Froude number on the dynamic head component?

Thanks for the comment. The following text will be added to the revised paper:

*The results showed that the Froude number ($Fr = \frac{v_f}{\sqrt{gD_w}}$) plays a major role in the relative contribution of the dynamic head coefficient; the higher the Froude number, the larger the dynamic head contribution (Figure 6a). The reason for this is that the amplitude of the hyporheic dynamic head depends on the ratio of the squared stream flow's velocity to its depth (Equation 6). The exact ratio varies with the relative penetration of bedforms ($Z_{BM}/D_w$), but generally, increasing the flow velocity and/or decreasing the flow depth leads to an increased contribution of the dynamic head component in the total hyporheic head.*

- L348-350: Why you mention this? Is it not expected?

In this section, we show the difference between the travel times of intermediate and deep groundwater (in the absence of hyporheic flow fields) in streambed sediment, and ultimately quantitatively assess the impact of hyporheic fluxes (i.e., in the presence of hyporheic flow fields) on said travel times (Figure 7). Deep groundwater moves slower than shallow groundwater, which is expected; however, here, we quantitatively show how the hyporheic flow affects the groundwater flow's travel times in streambed sediments.

- L495: In the long time, changes of the Surface and streambed morphology is expected, therefore it is not clear how these changes interfere with the flow field in the deeper deposits (Quaternary and bedrock).

Thanks for the comment. As mentioned earlier in response to a comment of the referee, a separate discussion on the dynamic behavior of the geological formation and topographical changes will be added to the revised paper.

- L510 and following lines: Die you take into account structural aspects of the heterogeneity in the bedrock, such as heterogeneity due to shear zones , mylonites etc.

We agree that this is a point that could be discussed more thoroughly, which we have tried to do in the revised version of the paper. Our study accounts for the horizontal heterogeneity of soil hydraulic conductivity, as well as the vertical stratification and depth-decaying hydraulic conductivities, which we think represents the most essential heterogeneity pattern in the absence of more detailed data. The heterogeneity of the bedrock domain was characterized by applying a depth-decaying hydraulic conductivity function. In addition, previous studies showed the significant effects of topography variation, as opposed to heterogeneity, in the fracture network within the bedrock domain on deep groundwater travel time in different subsurface strata, as well as on their discharge zones at the topographical surface (Selroos et al., 2002; Marklund et al., 2008; Welch et al., 2012). The following discussion is added to the revised paper to address the referee's comment:

> *Previous studies have indicated that the deep groundwater discharge location at the topographical surface is primarily controlled by topographical variation, and the distribution of the fracture network only slightly affects the discharge locations (Selroos et al., 2002; Marklund et al., 2008; Welch et al., 2012). In the absence of detailed data for the study region, the bedrock was assumed to be a continuum subsurface stratum with average hydraulic conductivity, and the horizontally constant hydraulic conductivity was assumed to decay with depth (Figure 1). In the present study, the applied depth-decaying hydraulic conductivity function was based on regional hydraulic conductivity field measurements conducted by Swedish Nuclear Fuel and Waste Management, SKB (Ericsson et al., 2006). Horizontal variability in the hydraulic conductivity was assumed, but only for the overlying soil strata (Quaternary deposits and possible aquatic sediments). These uppermost layers have a dominating effect on the discharge of groundwater, and thus, we believe that this simplified model represents the most important patterns in hydraulic conductivity.*

- Integrate Figures 1 & 2

Figure 1 and Figure 2 will be combined in the revised paper.

- Figure 4: Up-date: Northing? Points instead of stars?

The symbol for north will be changed and the deep groundwater discharge locations will be presented by points in the revised paper.

- Figure 9: Correct catchemnt

The word "catchment" will be corrected in the revised paper.

**References**

Ameli, A., McDonnell, J., and Bishop, K.: The exponential decline in saturated hydraulic conductivity with depth: a novel method for exploring its effect on water flow paths and transit time distribution, Hydrological Processes, 30, 2438-2450, 2016a.

Ameli, A., Amvrosiadi, N., Grabs, T., Laudon, H., Creed, I., McDonnell, J., and Bishop, K.: Hillslope permeability architecture controls on subsurface transit time distribution and flow paths, Journal of Hydrology, 543, 17-30, 2016b.

Dams, J., Salvadore, E., Daele, T. V., Ntegeka, V., Willems, P., and Batelaan, O.: Spatio-temporal impact of climate change on the groundwater system, Hydrology and Earth System Sciences, 16, 1517-1531, 2012.

Ericsson, L. O., Holmen, J., Rhen, I., and Blomquist, N.: Supra regional ground water modelling-in-depth analysis of the groundwater flow patterns in eastern Smaaland. Comparison with different conceptual descriptions, Swedish Nuclear Fuel and Waste Management Co., File Rep. SKB-R-06-64, 2006.

Genel, S., Vogelsberger, M., Nelson, D., Sijacki, D., Springel, V., and Hernquist, L.: Following the flow: tracer particles in astrophysical fluid simulations, Monthly Notices of the Royal Astronomical Society, 435, 1426-1442, 2013.

Grant, S. B., Stolzenbach, K., Azizian, M., Stewardson, M. J., Boano, F., and Bardini, L.: First-order contaminant removal in the hyporheic zone of streams: Physical insights from a simple analytical model, Environmental science & technology, 48, 11369-11378, 2014.

Ingebritsen, S. and Manning, C. E.: Geological implications of a permeability-depth curve for the continental crust, Geology, 27, 1107-1110, 1999

Jiang, X. W., Wan, L., Wang, X. S., Ge, S., and Liu, J.: Effect of exponential decay in hydraulic conductivity with depth on regional groundwater flow, Geophysical Research Letters, 36, 2009.

Jiang, X. W., Wan, L., Cardenas, M. B., Ge, S., and Wang, X. S.: Simultaneous rejuvenation and aging of groundwater in basins due to depth-decaying hydraulic conductivity and porosity, Geophysical Research Letters, 37, 2010.

Jiang, X. W., Wang, X. S., Wan, L., and Ge, S.: An analytical study on stagnation points in nested flow systems in basins with depth-decaying hydraulic conductivity, Water Resources Research, 47, 2011.

Jutebring Sterte, E., Lidman, F., Lindborg, E., Sjöberg, Y., and Laudon, H.: How catchment characteristics influence hydrological pathways and travel times in a boreal landscape, Hydrology and Earth System Sciences, 25, 2133-2158, 2021.

Kautsky, U., Lindborg, T., and Valentin, J.: Humans and ecosystems over the coming millennia: Overview of a biosphere assessment of radioactive waste disposal in Sweden, Ambio, 42, 383-392, 2013.

Laudon, H., Taberman, I., Ågren, A., Futter, M., Ottosson-Löfvenius, M., and Bishop, K.: The Krycklan Catchment Study—A flagship infrastructure for hydrology, biogeochemistry, and climate research in the boreal landscape, Water Resources Research, 49, 7154-7158, 2013.

Lidman, F., Peralta-Tapia, A., Vesterlund, A., and Laudon, H.: 234U/238U in a boreal stream network—Relationship to hydrological events, groundwater and scale, Chemical Geology, 420, 240-250, 2016.

Lidmar-Bergström, K.: Long term morphotectonic evolution in Sweden, Geomorphology, 16, 33-59, 1996.

Lidmar-Bergström, K.: A long-term perspective on glacial erosion, Earth Surface Processes and Landforms: The Journal of the British Geomorphological Group, 22, 297-306, 1997.

Lyon, S. W., Grabs, T., Laudon, H., Bishop, K. H., and Seibert, J.: Variability of groundwater levels and total organic carbon in the riparian zone of a boreal catchment, Journal of Geophysical Research: Biogeosciences, 116, 2011.

Marklund, L., Wörman, A., Geier, J., Simic, E., and Dverstorp, B.: Impact of landscape topography and Quaternary overburden on the performance of a geological repository of nuclear waste, Nuclear technology, 163, 165-179, 2008.

Morén, I., Wörman, A., and Riml, J.: Design of remediation actions for nutrient mitigation in the hyporheic zone, Water Resources Research, 53, 8872-8899, 2017.

Ryan, R. J. and Boufadel, M. C.: Evaluation of streambed hydraulic conductivity heterogeneity in an urban watershed, Stochastic Environmental Research and Risk Assessment, 21, 309-316, 2007.

Saar, M. and Manga, M.: Depth dependence of permeability in the Oregon Cascades inferred from hydrogeologic, thermal, seismic, and magmatic modeling constraints, Journal of Geophysical Research: Solid Earth, 109, 2004.

Salehin, M., Packman, A. I., and Paradis, M.: Hyporheic exchange with heterogeneous streambeds: Laboratory experiments and modeling, Water Resources Research, 40, 2004.

Selroos, J.-O., Walker, D. D., Ström, A., Gylling, B., and Follin, S.: Comparison of alternative modelling approaches for groundwater flow in fractured rock, Journal of Hydrology, 257, 174-188, 2002.

Song, J., Chen, X., Cheng, C., Summerside, S., and Wen, F.: Effects of hyporheic processes on streambed vertical hydraulic conductivity in three rivers of Nebraska, Geophysical Research Letters, 34, 2007.

Stanford, J. A. and Ward, J.: An ecosystem perspective of alluvial rivers: connectivity and the hyporheic corridor, Journal of the North American Benthological Society, 12, 48-60, 1993.

Sterte, E. J., Johansson, E., Sjöberg, Y., Karlsen, R. H., and Laudon, H.: Groundwater-surface water interactions across scales in a boreal landscape investigated using a numerical modelling approach, Journal of Hydrology, 560, 184-201, 2018

Swedish, N. F.: Site description of Forsmark at completion of the site investigation phase. SDM-Site Forsmark, Swedish Nuclear Fuel and Waste Management Co., 2008.

Toth, J.: A theoretical analysis of groundwater flow in small drainage basins, Journal of geophysical research, 68, 4795-4812, 1963.

Welch, L. A., Allen, D. M., and Van Meerveld, H.: Topographic controls on deep groundwater contributions to mountain headwater streams and sensitivity to available recharge, Canadian Water Resources Journal/Revue canadienne des ressources hydriques, 37, 349-371, 2012.

Whitaker, S.: Flow in porous media I: A theoretical derivation of Darcy's law, Transport in porous media, 1, 3-25, 1986.

Wörman, A., Packman, A. I., Johansson, H., and Jonsson, K.: Effect of flow-induced exchange in hyporheic zones on longitudinal transport of solutes in streams and rivers, Water Resources Research, 38, 2-1-2-15, 2002.

Zijl, W.: Scale aspects of groundwater flow and transport systems, Hydrogeology Journal, 7, 139-150, 1999.

---

## Author Comment (AC3)

**Response to Comments from Referee #1**

**General Comments**

- This paper investigates how hyporheic flow redistributes regional discharge at the scale of riverbeds in a case study and discusses the implications for the fate and transport of radionuclides from deep nuclear waste depositories. The study involves a multiscale flow and transport modelling framework and a rather sophisticated analysis of model outputs.
  The main result (fragmentation of the regional discharge by hyporheic flow) is rather intuitive and could already be anticipated by looking at Tóth (1963) flow fields. This takes nothing away from the merit of the study, which achieved a proper demonstration and quantification of this phenomenon in a realistic example. Therefore, I recommend publication without any doubt.
  The main issue I found is that the paper lacks a number of explanations (see below), but I am sure the authors can improve on this aspect.

We are grateful to the referee for the positive feedback on the paper. The paper is consistent with the hierarchically nested groundwater flow cells first described by Tóth (1963). However, as an expansion of the work of Tóth (1963), the geological heterogeneity of the subsurface (soil type and stratification) as well as the difference in boundary conditions in re- and discharge zones were taken into account. In particular, different boundary conditions representing flowing surface water, groundwater table topography and recharge-controlled boundary conditions were applied in discharge and recharge areas of the top surface of the catchment-scale numerical model. This required a consideration of the interaction between the nested groundwater flow systems across spatial scales and variation in boundary conditions. A detailed reply to the comments of the referee is provided below. The paper will be revised after the online discussion, in which the changes are indicated by *underline text in italic*.

**Detailed Comments**

- L51-56: I am not sure that velocity is the most relevant indicator for what you are trying to convey here. In fact, all the flow paths end up having similar velocities when approaching discharge (Cardenas and Jiang, 2010; Zijl, 1999; Zlotnik et al., 2011). Instead, I would think that the ratio of hyporheic flow (i.e., its total flow rate) to that of regional flow is more relevant in this discussion.

Thanks for the comment. The section will be revised as follows:

*Values founds in the literature indicate that the ratio of regional groundwater flow to hyporheic exchange flow in the streambed sediment is in the range of $10^{-2}$-$10^{-4}$ (Bhaskar et al., 2012; Goderniaux et al., 2013; Gomez-Velez et al., 2014).*

- L59: I guess you mean "principal effects" and not "principle effects".

Thanks for the comment. That is correct. The term will be corrected in the revised paper.

- L62-64: The sentence is grammatically incorrect.

Thanks for the comment. The sentence will be revised as follows:

*We hypothesize that hyporheic fluxes significantly influence the discharge of deep groundwater into streambeds, reflecting the fact that hyporheic fluxes, in combination with other factors (such as landscape topography, geology, and climate), control groundwater discharge at the scale of the watershed.*

- L110: This section should be better put before the description of the models.

Following the referee's suggestion, the section will be placed at the beginning of the "Methodology" section (before the description of the models) in the revised version of the paper.

- L111-114: A situation map would be useful (I would suggest including Figure S1 here).

A map of the study's catchment (Figure S1 from previously submitted supporting information) will be added to this section.

- L129: This sentence is unclear. I guess you mean: the mean annual runoff estimated from the stream discharge measurements was set as the infiltration (please correct if needed).

Thanks for the comment. The sentence will be revised as the referee suggested.

- L129: Is it reasonable to neglect overland flow in your study area (you may be overestimating infiltration)?

Thanks for the comment. Basically, overland flow occurs when the rainfall intensity exceeds the hydraulic conductivity of the saturated soil. The mean annual runoff (i.e., 400 mm/year), estimated from the precipitation data of the Krycklan catchment, is $1.27 \times 10^{-8}$ (m/s), which is significantly lower than any plausible value of the hydraulic conductivity of the land surface in the boreal forested catchment. The following text will be added to the paper to address the referee's comment:

*The infiltration rates in boreal forested catchments are generally higher than the precipitation rate, especially when the yearly average precipitation is used (Diamond and Shanley, 2003; Laudon et al., 2007). Hence, the mean annual runoff estimated from precipitation data was set as the infiltration rate (i.e., 400 mm/y was used as the estimated runoff value, provided by ©Swedish Meteorological and Hydrological Institute, SMHI; whereas the remaining precipitation, 214 mm/y, was used as the evapotranspiration rate (Karlsen et al., 2016)).*

- L129: Can you indicate the calculated infiltration rate?

The estimated infiltration rate provided by ©Swedish Meteorological and Hydrological Institute (SMHI) will be added to the following sentence:

*The mean annual runoff estimated from the precipitation data was set as the infiltration (i.e., 400 mm/y, as provided by ©Swedish Meteorological and Hydrological Institute, SMHI; the remaining precipitation, 214 mm/y, was used as the evapotranspiration rate (Karlsen et al., 2016)).*

- L161-162: I suggest referring to Haitjema and Mitchell-Bruker (2005) in support of this sentence.

Thanks for the suggestion. The suggested reference will be added to this sentence.

- L162-164: I suggest referring to Bresciani et al. (2016a, 2016b) in support of this sentence.

Thanks for the suggestion. The suggested references will be added to this sentence.

- L166-173: This only makes sense if hydraulic head is specified and equal to the topography along the top boundary, but you just said above that you are using a recharge condition, so I am lost here.

Mojarrad et al. (2019) indicated that the topography-controlled boundary condition is predominant in the Krycklan catchment. Therefore, topography was used as the main constraint of the top boundary condition in this study. However, the boundary condition should not result in a higher infiltration rate than the excess precipitation and, thus, the infiltration rate must be considered as an additional constraint in recharge areas. One way of achieving this goal (satisfying the infiltration rate) is to only smooth the topography DEM resolution over the groundwater recharge zones to represent the recharge-controlled boundary condition (Marklund and Wörman, 2011; Wang et al., 2018), while maintaining the high resolution of the DEM file over the discharge areas to represent the topography controlled case. As a consequence, both the recharge- and topography-controlled boundary conditions were used in our groundwater model.

We have clarified the sentence,

"The applied method helps to satisfy both the Dirichlet and Neumann boundary conditions through a practical way of recognizing an unknown and spatially variable infiltration."

as follows in the manuscript:

*"The applied method implies that the numerical solution is formally derived using a constant head (Dirichlet boundary condition) at the groundwater table, but the smoothed boundary also satisfies the limited infiltration (Neumann boundary condition)."*

- L134-185: What are the horizontal limits of the domain?

The horizontal limits of the domain will be presented in the paper as follows:

*The bounding rectangle of the Krycklan catchment (i.e., 11.6 × 10.3 km²) was set as the horizontal limit of the numerical model's domain (64°11.8´N - 64°17.6´N, 19°39.5´E - 19°54.3´E).*

- L194: I do not understand the meaning of "$C_{damp}(\lambda_i)$". Is $C_{damp}$ a function of $\lambda_i$ (I would think not since $C_{damp}$ seems to be treated as a constant)? And if it is, shouldn't it be $\lambda_{ij}$?...

Thanks for the insightful comment. The results of previous studies (Morén et al., 2017; Mojarrad et al., 2019) show that the hydrostatic damping factor ($C_{damp}$) of the stream's surface, resulting from the independent spectral analysis conducted in 1D, is a function of wavelength ($\lambda$). In particular, the results of a field investigation conducted by Morén et al. (2017) indicate that the damping factor is close to 1 for topographic wavelengths greater than 300 m, and that it decreases to 0.3 for wavelengths of around 5 m. Therefore, the damping factor should have been written as $C_{damp}(\lambda_{i,j})$, but it only varies with index i, and considered as constant with index j. Equation 5 and the notation for the damping factor will be corrected throughout the paper. The following text will also be added to the revised version of the paper to address the referee's comment:

*$C_{damp}(\lambda)$ is the hydrostatic damping factor representing the smoothness of the water's surface in comparison to the streambed surface, which has been shown to be a function of wavelength (Morén et al., 2017). In the present study, $\lambda_{i,j}$ was used as the wavelength in the x and y directions, but it should be noted that the damping factor only varied with an index i, and was treated as a constant with index j.*

- L206-216: How does this relate to the previous paragraph?

The high-resolution streambed topography data required for hyporheic modeling were not available. However, the fractal pattern of the topography's fluctuation allows us to rescale the landscape topography to the smaller streambed scale. As such, the spectral solution to groundwater hydraulic head fluctuation was described in L195-205. Then, L206-216 describes the justification and the method for the rescaling of landscape topography (which was estimated in L195-206), in order to evaluate the streambed's hydraulic head at the hyporheic scale.

The following is the relevant section of the paper, where the added/revised parts are shown in italic and underlined font.

*The topography of the landscape and the streambeds* have been shown to follow fractal patterns, allowing a spectral representation of the head boundary condition, as well as solutions to topography-controlled groundwater circulation (Wörman et al., 2006, 2007b). The fractality reflects a constant power law correlation between the topographic amplitude and the wavelength across all scales in a real Fourier series representing the topographic elevation. This fractal power has been shown to prevail over a wide range of scales, from continents to bedforms in streams (Wörman et al., 2007a), suggesting the possibility of generalizing ground surface topography, such as streambed topography, over scales *for which the high-resolution streambed topography data required for hyporheic modeling are not available*.

- L187-216: What are the extent and boundary conditions of the hyporheic flow model? I guess the boundary conditions must be head = 0 everywhere but the top boundary so as to keep a continuous solution when doing the superposition...?

Thanks for the insightful comment. We have revised this sentence on line 187,

"The streambed hydraulic head was applied as a boundary condition for the hyporheic flow,..."

in the following way:

*"In order to superimpose the results of the analyses of hyporheic flow onto the regional groundwater flow, at the top-boundary of the hyporheic flow domain, we only recognized the local fluctuations of the streambed hydraulic head from the regional hydraulic head. Hence, the fluctuations of the streambed hydraulic head were applied as a boundary condition for the hyporheic flow, i.e., the flow of surface water through streambed sediment inflow paths that re-emerges into surface water. The hyporheic flow model was analyzed at many points where discharge from deep groundwater was found (see section 2.5). These areas were used to determine the effect of the hyporheic flow on the discharge of groundwater."*

In addition, the hyporheic scale model was represented with $5 \times 5 \times 5$ m$^3$ cubes, wherein the hydraulic head estimated using the spectral exact solution (Equation 5 of the paper) was used as the top boundary condition. In addition, no flow boundary was assumed for the bottom and lateral surfaces of the hyporheic model. The hydraulic conductivity of the hyporheic-scale model was described according to Equation 4 (i.e., similarly to the streambed sediment layer of the regional-scale model). The following sentences will be added to the paper to address the referee's comment:

*Finally, the hyporheic-scale mode was represented with $5 \times 5 \times 5$ m$^3$ cubes, wherein the hydraulic head was estimated using the spectral exact solution (Equation 5) and was used as the top boundary condition. In addition, no flow boundary was assumed for the bottom and lateral surfaces of the hyporheic model. The hydraulic conductivity of the hyporheic-scale model was determined via to Equation 4 (i.e., similar to the streambed sediment layer of the regional-scale model).*

- L219: The term "models" is confusing here. I guess you refer to the other parameters of the hyporheic flow model and all the parameters of the catchment-scale model.

Thanks for the comment. The sentence will be revised as follows:

*This study recognized uncertainties in the hydrostatic and dynamic head boundary conditions by performing a sensitivity analysis on the parameters in Equation (5), while the uncertainty in the other parameters of the hyporheic flow model (such as hydraulic conductivity, etc.) and in those of the catchment-scale flow model was not formally analyzed.*

L239: "Carlo", not "Carla".

Thanks for the comment. This will be corrected in the revised version of the paper.

- L247: What are "the" cubes? You have not talked about cubes before.

Thanks for the comment. A new section will be added to the revised version of the paper at the beginning of section 2.6 to describe the cubes.

- L252: What does "the 1552" refer to?

It refers to discharge points within the catchment boundaries. The text will be revised as follows:

*The catchment-scale groundwater velocity field was superimposed on the corresponding streambed-scale velocity field (i.e., the corresponding deep groundwater discharge zone at each of the 1552 discharge points within the catchment boundaries; see section 3.1 and figure 4).*

- L253-256: How did you distinguish between intermediate and deep groundwater flow paths from these particles (did you track them backward as well)? Furthermore, how can you be sure that some of these particles are not hyporheic flow?

Thanks for the comment. Particle tracking was used to identify streamlines that are considered to follow hyporheic, intermediate and deep groundwater flow. Backward particle tracking was conducted to distinguish the deep and intermediate groundwater flow paths, where deep groundwater was defined as flow that entered the bedrock and intermediate groundwater was determined as being only present in Quaternary deposits. Hyporheic flow involves streamlines starting and ending in the stream's bottom within the 5 m spatial scale. No flow boundary condition was assumed for the bottom surface of the hyporheic model (i.e., the $5 \times 5 \times 5$ m$^3$ cubes), which allowed us to analyze the hyporheic flow without considering the ambient groundwater discharge. Therefore, none of the hyporheic flow paths originate from/touch the bottom surfaces of the superimposed models.

Particle tracing in the superimposed models was conducted by releasing particles at the bottom of the domain of each superimposed flow field (i.e., at a depth of 5 m). As such, the released particles reflected only the deep and intermediate flows in the superimposed models (Figure 2 illustrates the applied method).

A separate section will be added to the revised version of the paper to describe the method of particle tracking for different flow types, as follows:

> *Particle tracking was used to identify streamlines that follow hyporheic, intermediate and deep groundwater flows. Deep groundwater was defined as flow that entered the bedrock, and intermediate groundwater was determined to only be present in Quaternary deposits. Hyporheic flow contains streamlines starting and ending in the stream bottom within a 5 m spatial scale. Particle tracking was conducted at two different spatial scales: the regional scale (i.e., entire catchment) and the local scale (i.e., 5×5×5 m³). The regional-scale particle tracking was conducted to evaluate deep groundwater discharge zones, as well as to distinguish the deep and intermediate groundwater flow fields (see section 2.6.1). In addition, particle tracking was conducted on a large number of 5×5×5 m³ cubes containing a hyporheic flow field in deep groundwater discharge zones, in the absence and presence of upwelling groundwater (both deep and intermediate flows), in order to investigate the impact of hyporheic flow on deep groundwater discharge.*

As was mentioned in response to the referee's comment, a new description of the boundary conditions applied to hyporheic-scale flow will be added to the revised version of the paper (section 2.4) so as to facilitate the understanding of the nested flow system, as follows:

> *Finally, the hyporheic-scale model was represented by 5 × 5 × 5 m³ cubes, wherein the hydraulic head was estimated using the spectral exact solution (equation 5) and was used as the top boundary condition. In addition, no flow boundary was assumed for the bottom and lateral surfaces of the hyporheic model. The hydraulic conductivity of the hyporheic-scale model was determined via Equation 4 (i.e., similar to the streambed sediment layer of the regional-scale model).*

- L264-272: This part could be clearer. Did you focus on the same 1552 areas as above (I guess so)? How did you determine the coherent catchment-scale discharge areas (I guess this would involve particle tracking and a certain grouping method)?

We agree with the referee that this part could be explained better. The coherent areas were evaluated in 5×5×5 m³ cubes of the superimposed models, with and without the hyporheic flow field, in order to assess the impact of hyporheic flux on catchment-scale groundwater flow in these areas. In addition, the fragmentation analysis was conducted on catchment-scale model's top surface (for the whole catchment) to present the size distribution of the coherent groundwater upwelling zones throughout the whole catchment, regardless of groundwater penetration depth (i.e., no particle tracing was involved in the fragmentation of coherent upwelling zones across the catchment).

The following is the relevant section of the paper, in which the added parts are presented in italic and underlined font.

> In addition to the deep groundwater travel time in the superimposed models, the analyses also covered how the hyporheic flows affected the spatial distribution of various sizes of catchment-scale groundwater upwelling zones at streambed interfaces. *The fragmentation analysis was conducted on deep groundwater flow discharge zones (i.e., 1552 discharge zones within the catchment boundaries; refer to section 3.1 and Figure 4) using the particle tracing results for the 5×5×5 m³ superimposed cubes.* These results were used to determine the fragmentation of catchment-scale groundwater flows arising at streambed interfaces, as defined from the change in the distributions of coherent areas that only experienced the upwelling of catchment-scale groundwater flow. The changes in coherent discharge areas were determined by superimposing and not superimposing the hyporheic flows on the catchment-scale groundwater flows. In this study, a coherent area was defined as an area in which the entire flow reflected only the catchment-scale upward groundwater flow. *In addition, fragmentation analysis was conducted on the catchment-scale model's top surface (with a resolution of 5×5 m² in different locations over the whole catchment) to determine the size distribution of coherent upwelling zones regardless of groundwater penetration depth.* Numerically, coherent upwelling areas were evaluated at the top surfaces of the streambed-scale and catchment-scale models using an orthogonal mesh with resolutions of 0.1×0.1 m² and 5×5 m², respectively, wherein the flow velocity values were considered only in the orthogonal directions.

- L308-310: I think the differences between the three layers are mostly independent of the hierarchical structure of flow cells (which was not evaluated in this study, by the way).

We agree with the referee that our paper evaluated differences in flow properties (i.e., travel time, velocity, etc.) between the three layers due to variations in the porosity, hydraulic conductivity, and thickness of those three subsurface layers. However, previous studies (Cardenas, 2007; Wang et al., 2016) have indicated that differences in the distribution of groundwater travel time are due to the hierarchical structures of groundwater flow cells. References will be added to this section of the paper to support the argument and to address the referee's comment (i.e., Cardenas, 2007; Wang et al., 2016).

- L331: Define the Froude number.

The Froude number will be defined in the revised version of the paper, as follows:

_The results showed that the Froude number ($Fr = \frac{v_f}{\sqrt{g D_w}}$) plays a major role in the relative contribution of the dynamic head coefficient; the higher the Froude number, the larger the dynamic head contribution (Figure 6a)._

- L418-420: So is it a good news (less exposure time of aquatic sediments)?

Yes, the velocity of the groundwater flow (carrying, for example, radionuclide compounds from deep groundwater) is increased due to the presence of the hyporheic flow field; on the other hand, though, the groundwater discharge area at the sediment bed interface shrinks due to the impact of hyporheic flux. This reflects the shorter exposure time (due to the higher velocity) and the higher radionuclide activity (due to the smaller discharge area).

- L421: Why would it lead to higher exposure if the exposure time is shorter?

Thanks for the comment. As mentioned in the response to a previous comment, very narrow pinhole discharge areas of deep groundwater at the sediment bed interface result in higher radiologic activity (not higher exposure time). The sentence will be revised in the paper as follows:

_Moreover, hyporheic flow causes the upwelling of deep groundwater to become more spatially focused, and also causes the accumulation of any radioactivity that may follow the flow in small areas, which potentially causes greater radiologic activity. The shorter residence time could result in reduced exposure time, but for most radionuclide compounds, these times would be much longer than the life-span of humans. Further, a lengthy duration of leakage (from a damaged radionuclide waste repository) might determine the actual period of contamination of aquatic sediments, thus reducing the importance of the residence time in those sediments._

**References**

Bhaskar, A. S., Harvey, J. W., and Henry, E. J.: Resolving hyporheic and groundwater components of streambed water flux using heat as a tracer, Water Resources Research, 48, 2012.

Bresciani, E., Gleeson, T., Goderniaux, P., de Dreuzy, J.-R., Werner, A., Wörman, A., Zijl, W., and Batelaan, O.: Groundwater flow systems theory: research challenges beyond the specified-head top boundary condition, Hydrogeology Journal, 24, 1087-1090, 2016a.

Bresciani, E., Goderniaux, P., and Batelaan, O.: Hydrogeological controls of water table-land surface interactions, Geophysical Research Letters, 43, 9653-9661, 2016b.

Cardenas, M. B.: Potential contribution of topography-driven regional groundwater flow to fractal stream chemistry: Residence time distribution analysis of Tóth flow, Geophysical research letters, 34, 2007.

Cardenas, M. B. and Jiang, X. W.: Groundwater flow, transport, and residence times through topography-driven basins with exponentially decreasing permeability and porosity, Water Resources Research, 46, 2010.

Diamond, J. and Shanley, T.: Infiltration rate assessment of some major soils, Irish Geography, 36, 32-46, 2003

Goderniaux, P., Davy, P., Bresciani, E., de Dreuzy, J. R., and Le Borgne, T.: Partitioning a regional groundwater flow system into shallow local and deep regional flow compartments, Water Resources Research, 49, 2274-2286, 2013.

Gomez-Velez, J. D., Krause, S., and Wilson, J. L.: Effect of low-permeability layers on spatial patterns of hyporheic exchange and groundwater upwelling, Water Resources Research, 50, 5196-5215, 2014.

Haitjema, H. M. and Mitchell-Bruker, S.: Are water tables a subdued replica of the topography?, Groundwater, 43, 781-786, 2005

Karlsen, R. H., Grabs, T., Bishop, K., Buffam, I., Laudon, H., and Seibert, J.: Landscape controls on spatiotemporal discharge variability in a boreal catchment, Water Resources Research, 52, 6541-6556, 2016.

Laudon, H., Sjöblom, V., Buffam, I., Seibert, J., and Mörth, M.: The role of catchment scale and landscape characteristics for runoff generation of boreal streams, Journal of Hydrology, 344, 198-209, 2007.

Marklund, L. and Wörman, A.: The use of spectral analysis-based exact solutions to characterize topography-controlled groundwater flow, Hydrogeology Journal, 19, 1531-1543, 2011.

Mojarrad, B. B., Riml, J., Wörman, A., and Laudon, H.: Fragmentation of the hyporheic zone due to regional groundwater circulation, Water resources research, 55, 1242-1262, 2019.

Morén, I., Wörman, A., and Riml, J.: Design of remediation actions for nutrient mitigation in the hyporheic zone, Water Resources Research, 53, 8872-8899, 2017.

Toth, J.: A theoretical analysis of groundwater flow in small drainage basins, Journal of geophysical research, 68, 4795-4812, 1963.

Wang, J.-Z., Wörman, A., Bresciani, E., Wan, L., Wang, X.-S., and Jiang, X.-W.: On the use of late-time peaks of residence time distributions for the characterization of hierarchically nested groundwater flow systems, Journal of Hydrology, 543, 47-58, 2016.

Wang, C., Gomez-Velez, J. D., and Wilson, J. L.: The importance of capturing topographic features for modeling groundwater flow and transport in mountainous watersheds, Water Resources Research, 54, 10,313-310,338, 2018.

Wörman, A., Packman, A. I., Marklund, L., Harvey, J. W., and Stone, S. H.: Exact three-dimensional spectral solution to surface-groundwater interactions with arbitrary surface topography, Geophysical research letters, 33, 2006.

Wörman, A., Marklund, L., Xu, S., and Dverstorp, B.: Impact of repository depth on residence times for leaking radionuclides in land-based surface water, Acta Geophysica, 55, 73-84, 2007a.

Wörman, A., Packman, A. I., Marklund, L., Harvey, J. W., and Stone, S. H.: Fractal topography and subsurface water flows from fluvial bedforms to the continental shield, Geophysical Research Letters, 34, 2007b.

Zijl, W.: Scale aspects of groundwater flow and transport systems, Hydrogeology Journal, 7, 139-150, 1999.

Zlotnik, V. A., Cardenas, M. B., and Toundykov, D.: Effects of multiscale anisotropy on basin and hyporheic groundwater flow, GroundWater, 49, 576-583, 2011.

---

## Author Comment (AC4)

**Response to Community Comments #2 by: Ulrik Kautsky**

- We had an additional important comment to the manuscript lost in my previous post.
  Many of the claims regarding ecology and human health are outside of the scope of the study especially given the simplifications that have been made regarding the hydrology of the biosphere. An attempt to extrapolate results into these areas is not properly motivated. Results of the study will stand for themselves without having to extrapolate them into areas outside of the study's scope which, at present, is largely theoretical.

We thank Ulrik Kautsky for the comment. Our aim is not to specifically claim what implications the results might have for the understanding of the impact of radionuclide migration on human health and ecology. The paper indicates that there is potentially a relationship between the results and the rate coefficients of radionuclide transport models used for the biosphere assessment, which, in principle, would have wider implications. Further, the contaminated area distribution is a factor for transport modeling. Therefore, we revised the parts of the paper describing the implications of the results for ecology and human health, in order to avoid any misunderstanding.

Lines 422-424 will be revised as:

*Hence, the results from models of radionuclides transported by deep groundwater flow within the streambed sediment need to be considered as a basis for transport modeling in the biosphere, and for assessing doses to humans.*

Lines 56-58 will be revised as:

*In particular, the residence times of radionuclides in streambeds and the sizes of regional groundwater discharge areas may be affected by hyporheic flows, which could be used as the input in dose assessment models to investigate radionuclide's spread in surface water over long periods.*

Lines 452-454 will be revised as follows:

*Hence, a higher retardation factor in Quaternary deposits and sediments compared to that in bedrock leads to the accumulation of radionuclides within aquatic sediment for long periods of time, which needs to be considered in dose assessment models concerning the potential prolonged exposure of humans to radiological doses.*

---

## Author Comment (AC5)

**Response to Comments from Referee #3**

- The authors of this study used a complex multiscale model to compare the travel time and spatial distribution of catchment scale groundwater flow with and without the influence of hyporheic flow. The results are loosely linked to the fate of nuclear particles that might potentially leak out from nuclear waste disposal in the bedrock of the catchment. The main finding of the study is the fragmentation of the groundwater discharge areas, the reduction of the effective volume that carries groundwater flow and the consequential reduction of the groundwater travel time in the upper 5m of the domain. A monte carlo approach was used to some extend to generate a distribution of possible outcomes where exact model parameters were unknown.

  The underlying model is without a question quite sophisticated. However, I wonder if the right model was chosen to answer the specific scientific question. The main finding, that the discharge zone is fragmented, is relatively obvious and has been published various times in model studies from the hyporheic perspective (e.g. Boano et al 2008, Trauth et al 2014, Fox et al 2016) and also investigated experimentally (e.g. Bhaskar 2012). Given that the general effect is well known, I would have expected an in-depth analysis of the correlations of at least some of the parameters involved. I wonder if this complex, multiscale, pseudo-realistic model is the right choice to draw systematic conclusions with results that have a direct use for other scientists. The model has too many unknowns to investigate general findings, e.g. correlating discharge area reduction to the ratio of hyporheic head and groundwater head or something similar.

The authors appreciate the referee for taking the time to review and comment on the paper. However, we believe that the referee might have misunderstood the applied methods, which combine a regional flow model and a local, hyporheic flow model. The focus is on the effect of hyporheic flow on the discharge of groundwater, not vice versa. Thus, there are major differences in the method and focus of this study compared to previous research, which are highlighted in this response and in the manuscript. For example, the papers of Trauth et al. (2014), Fox et al. (2016) and Bhaskar (2012) deal with ambient groundwater flow, but this is not spatially variable in a watershed. The paper of Boano et al. (2008) deals with regional groundwater circulation, but does not focus on the variability in groundwater discharge. We have tried to stress these differences more clearly in the introduction of the paper.

In summary, the main differences are as follow: (a) previous studies have investigated the impact of upwelling groundwater on the hyporheic flow field and the discharge of groundwater, without considering the hyporheic flow; (b) the suggested research that addressed hyporheic flow did not investigate the "fragmentation of discharge zones" and how the discharge areas/lengths are affected by multiscale subsurface flow interactions; (c) none of the suggested papers evaluated the hierarchically nested structure of the deep groundwater flow on a regional scale, and therefore, (d) the heterogeneous patterns of groundwater gaining and losing reach are not represented on the landscape-scale in the papers suggested by the referee.

Considering all the aforementioned factors, we believe that our results elucidate a phenomenon that is not addressed in previous research.

The following text will be added to the Introduction of the revised paper in order to highlight the focus of the research:

> *Previous studies have investigated the effects of upwelling groundwater on hyporheic flow fields (e.g., Bhaskar et al., 2003; Trauth et al., 2014; Fox et al., 2016), which fundamentally differs from the focus of this paper. This paper investigates the impact of hyporheic flow on upwelling groundwater. In addition, the hierarchically nested structure of the regional scale groundwater was not recognized in previous research on hyporheic exchange flow (e.g., Boano et al., 2008). The aim of the present study is to investigate the principal effects of hyporheic flow on the discharge of groundwater originating from depths much greater than those of hyporheic flow.*

The model's complexity and the need to support it with field data are indeed relevant issues, especially for this complex problem. The investigated flow phenomena require a modeling framework with a sufficient degree of complexity for both stream flow and regional groundwater conditions, in order to evaluate the subsurface nested flow system. The conceptual methodological framework used here allows us to address the subsurface flow interaction taking place on multiple scales within the streambed sediment in a proper and comprehensive (though not too detailed) format, which we believe could not be done more simply than in the presented research. The

model's framework can probably be simplified; however, this study did not thoroughly address the optimal level of complexity for the model, and instead used sensitivity analysis to address certain aspects.

- If the authors didn't aim to draw general results but rather calculate the effect for a specific case (nuclear waste disposal in this specific catchment), I see two problems. First, case studies do not match the scope of the HESS journal. Second, some of the boundary conditions have been selected without verification or are too unclear to be used for a specific use case (especially a safety relevant topic like nuclear waste disposal). Generally, I think that a lot of choices for boundary conditions are justified only by references to past work of the own workgroup. In some cases, references to independent workgroups would strengthen the trustworthiness of the model (see detailed comments below).

The aim of the study is described in the following manner:
"The aim of the present study is to investigate the principle effects of hyporheic flow on the discharge of groundwater originating from depths much larger than hyporheic flow depths. The general approach was, thus, to decouple the subsurface flow processes and analyze the catchment-scale groundwater flow field and its associated discharge zones with and without consideration of hyporheic flows. We hypothesize that hyporheic fluxes substantially influence deep groundwater discharge into streambeds and that hyporheic fluxes, in combination with other factors (such as landscape topography, geology, and climate), controls groundwater flows at the watershed scale. In the present study, a multiscale modeling framework was established to represent the spatial scales of both the geomorphology of a region and its subsurface flow field; this model was then applied to a boreal watershed in Sweden."

Hence, the study aim was to provide a more comprehensive picture of the investigated phenomenon, rather than a case-specific investigation.

We agree with the referee regarding the uncertainty of the applied boundary conditions, but maintain that the top-boundary condition is the most important for the posed problem. Other boundaries are placed far enough away to not significantly influence the flow within the watershed discharge zone. The lower boundary was moved to a sufficient depth to not influence the discharge zones, and the same applies for the location of the vertical delimitation of the flow domain. The top-boundary condition is a combination of topography and recharge-controlled boundary condition, both of which are well-established in geo-hydraulics. Both are supported by decent field data.

- The second major finding is the reduced travel time within the upper 5m of the 500m deep domain. Also here, I miss a systematic investigation of the underlying mechanism. E.g. what is the correlation between the fragmentation and the travel time? What is the correlation with the hyporheic head and groundwater head? Is the conductivity a relevant factor in this correlation? These kind of questions should be answered with specific results rather than vague discussions, because the vague answers to these questions are obvious. Again, it would have been much easier to answer these kind of questions if the model was less complex.

The paper's results on the impact of hyporheic flow fields were quantitatively presented, both in terms of the CDF of the discharge areas (Figure 9) and the travel times of the groundwater flow (Figure 7). Although the correlation between the fragmentation of the discharge zones and the flow travel time was not directly investigated in this paper, Figures 7 and 9 help us to understand this correlation in the presence of a hyporheic flow field. Therefore, we will modify the existing discussion on the impact of hyporheic fluxes to address the comment of the referee in the revised version of the paper, as follows;

*The correlation between the fragmentation of discharge zones and the flow travel time was not directly investigated in this study, but the reductions in travel time of both intermediate and deep groundwater flow (Figure 7), and in the discharge areas of the groundwater flow in the streambed sediment (Figure 9) in the presence of a hyporheic flow field, suggest an inverse correlation between fragmentation and the travel time of the flow. This means that the greater the fragmentation, the lower the groundwater flow travel time. This finding could be expected due to flow continuity, whereby the area and the time are directly correlated for a constant flow tube and for approximately the same flow trajectory. Thus, as the groundwater flow converges in smaller areas due to the streambed-induced fluxes, the flow velocities increase for both the intermediate and the deep groundwater. The travel time of the flow has an inverse relation with the flow velocity, meaning that higher velocities reduce the flow travel times of aquatic sediments (Figure 7).*

The following discussion will be added to the revised paper to address the correlation between hyporheic and regional groundwater hydraulic heads.

*The stream bedform variations that governed the hyporheic flows were randomly selected from the entire catchment, then rescaled and used in a Monte Carlo simulation. Therefore, the hyporheic topographical factors are assumed to be independent of their location in the catchment, and thus independent of the regional groundwater flow.*

Further, the following discussion will be added to the revised version of the paper to address the effect of hydraulic conductivity on the intensity of the hyporheic and deep groundwater flow velocities within the streambed sediment.

*Hydraulic conductivity is a primary controlling factor for both hyporheic and regional groundwater flow, due to its linear role in the Darcy law; thus, it causes a correlation in terms of the intensity of the two flow types. The regional groundwater flow, however, is controlled by hydraulic conductivity over larger distances, and generally the flow intensity is dominated by the lower values of hydraulic conductivity (i.e., those found in bedrock). This is because the average hydraulic conductivity over a flow path is given by the harmonic mean weighted by the distance of the path.*

- In addition to the missing universal results, I don't understand why the reduction of travel time in the upper 5m is deemed relevant. The effect of the hyporheic zone decreases exponentially with depth (see eq(5) or Elliot & Brooks 1997a), which is why only the very last segment of a particles travel path will be influenced. In Line 353 it says that the effect of the hyporheic head is strongest, where the overall seepage velocity is low. That means that particles that traveled centuries to millennia (Fig 5) to reach the surface will lose a few years on their last few meters (Fig 7). Even when the different retention coefficients in bedrock and deposits were considered (missing in Fig 7), the effect should be minor. I miss a clear explanation what processes are potentially influenced by the change in travel time in this last stage of the streampath.
  Overall, I'm afraid that the study design is not able to answer the research questions to a degree that goes beyond the intuitive and well-published findings.

It is correct that the contribution of transport in the near stream environment to the overall residence time of the water in the entire sub-surface flow path, starting at great depth in the bedrock, is small. However, this paper aims to highlight the effect of hyporheic fluxes on discharge zones, since it is in this environment that deep groundwater interacts most significantly with stream ecosystems, especially in terms of heat and solute fluxes transported via discharging water. The relatively thin Quaternary deposits of the area make this environment even more important. This research can find technical application in safety assessments of the geological disposal of high-level radioactive spent nuclear fuel, which includes scenarios of leakage where radionuclides flow into groundwater. As revealed by this study, the prolonged travel time of radionuclides in aquatic sediments exposes ecosystems and poses a threat to exposed humans. Although the ecological effects of the prolonged travel times exceed the scope of this study, an improved understanding of the flow process and the spatial distribution of ground water discharge zones will be an important aspect of dose assessment models at the geosphere–biosphere interface. Therefore, even though the water has traveled for millions of years before reaching the sediment, many important environmental processes occur in shallow groundwater zones.
Lines 405-424 of the submitted paper discuss this issue.

**Detailed Comments**

- L102: I don't think it is necessary to show the definition of Darcy's velocity, but if you do, please use the already defined symbol for Darcy's velocity "W_c" instead of q and q_seepage.

$W_C$ is the vertical component of Darcy velocity, while the vector quantity **q** refers to Darcy's flow velocity (**q** = [U,V,W]). In addition, one of the comments of the second referee concerns the definition of Darcy's velocity. Therefore, we will add the following definition of Darcy's velocity into the revised version of the paper in order to address the first and the second referees' comments:

*The momentum equation for subsurface flow in saturated porous media at a sufficiently low Reynolds number is generally derived by neglecting inertia terms, while maintaining the potential energy and adopting a linear friction-loss relationship. This leads to the well-known Darcy's law: $q = [U, V, W] = -K\nabla H$. Here, K (m/s) is the hydraulic conductivity, $\nabla$ is the nabla operator, H (m) is the total hydraulic head, q (m/s) is the Darcy*

*velocity vector, and U (m/s), V (m/s) and W (m/s) are the Darcy's velocities in the x, y, and z directions, respectively, while bold symbols denote vector quantities (Whitaker, 1986).*

- L108: The 50-fold difference between retardation factors between bedrock and deposit is a dominating factor for the distribution of particle travel times. It could be justified by a reference to other workgroups who found similar retardation factor ratios between rock and deposit.

Thanks for the comment. Two references (Jakubick, 1979; Neretnieks, 1979) will be added in the revised version of the paper to substantiate the values employed for the retardation factors in different subsurface media in this study.

- L147: Extrapolating a decay coefficient from measurements at 3 and 7 cm to a depth of 5 m is questionable. The assumption that the minimum conductivity is $10^{-6}$ (m/s) probably determines the decay coefficient much more than the actual measurements. The only reference is, again, only a single study from the own workgroup. The authors should be able to find more measurements of sediment conductivities in the literature to strengthen their assumption.

Thanks for the comment. In the revised version, we support our assumption (i.e., the depth-decaying hydraulic conductivity of the streambed sediment) by referring to field investigation studies conducted by other research groups (Ryan and Boufadel, 2006; Song et al., 2007; Singh et al., 2014), and we will also add the following to the Discussion of the revised paper.

*The depth-decaying hydraulic conductivity behavior of streambed sediment has been shown in previous research using field-measured data (Ryan and Boufadel, 2006; Song et al., 2007). These studies generally confirm the presence of depth decay in the hydraulic conductivity, which is more pronounced at shallow depths and then decreases with depth similarly to an exponential function (Singh et al., 2014). Therefore, in this study, we recognized the depth-decaying trend in hydraulic conductivity, and the exponential decay function was fitted to the data measured by Morén et al. (2017). Further, one can consider how grain size distributions and soil porosity vary with depth, and theoretically argue that these trends would affect the hydraulic conductivity that supports the selection of the functional type. However, it is most important here to represent the observed ratio of K-values in streambed sediments between the 3 cm and 7 cm depths, which varies from 10 to 90%. As a consequence, the epistemic (systematic) uncertainty included in the hyporheic flow fields, which is induced by the depth-decaying hydraulic conductivity function and the spatial heterogeneity of streambed sediments, was not recognized in this study.*

- L159 – 173: In line 165 you state that the realistic boundary condition would be a recharge-controlled boundary for most of the terrain. However, you choose a head boundary condition instead and use a mesh-coarsening algorithm to fit the recharge. Why didn't you simply use a recharge-controlled boundary? Coarsening the mesh to fit a result is somewhat unorthodox. All discretizing simulation techniques have in common that an infinitely fine mesh resolution results in the exact solution of the underlying differential equations. The boundary condition in this study, however, implies a tradeoff between model inaccuracy and boundary condition inaccuracy, which, in my opinion, should be avoided by choosing mesh-size independent boundary conditions.

Only the average recharge boundary condition is known for the region, whereas the landscape topography reflects the spatial distribution of the head boundary condition. In general, the top-boundary can be classified as either topography- or recharge-controlled, depending on the aquifer properties and climatic conditions of the study catchment (Gleeson et al., 2011). Haitjema and Mitchell-Bruker (2005) introduced a dimensionless water table ratio, WTR, to represent the connection between topography variation and the groundwater table, in which WTR>1 reflects the topography-controlled boundary condition, and WTR<1 represent the recharge-controlled boundary condition. Mojarrad et al. (2019) calculated the WTR for the Krycklan catchment and found the ratio to be higher than 1. Therefore, the landscape topography was set as the top boundary condition in discharge areas in this study. However, applying the topography-controlled boundary condition in recharge areas would result in too high infiltration rate. To resolve the constraints of the infiltration rate, we smoothed the landscape topography by decreasing the resolution of the mesh size (i.e., groundwater table) over the recharge areas (Marklund and Wörman, 2011; Wang et al., 2018).

- L177: A figure of the mesh/domain would be helpful. The domain is rectangular? I originally thought it was a whole (sub-)catchment with its natural boarders (and lateral no-flow boundary conditions).

Yes, the modeling domain is rectangular and hydrostatic pressure is applied along the side walls. This means that groundwater can flow through both the side walls of the domain, and the landscape water thus divides, as it should. The watershed is generally not defined by no-flow boundaries, due to the hierarchically nested character of the groundwater flow. An illustration of the domain will be added to the revised version of the supporting information. The details of the applied boundary conditions (bottom, top, and lateral surfaces) have already been described in lines 174-177. The geometric horizontal limits of the model are added in the revised version of the paper, as below:

*The bounding rectangle of the Krycklan catchment (i.e., 11.6 × 10.3 km²) was set as the horizontal limit of the numerical model domain (64°11.8´N -64°17.6´N, 19°39.5´E-19°54.3´E).*

In addition, the mesh size and the quality of the mesh elements will be included in the revised version of the paper, as follows:

*A nonuniform mesh was used for the flow calculation of each layer. The mesh sizes varied within the ranges of 0.1-2 m, 2-17 m and 17-403 m for the streambed sediment, Quaternary deposit and bedrock layers, respectively. In addition, the maximum element growth rate and the curvature factor for the streambed sediment are 1.15 and 0.1, while these change to 1.2 and 0.2 for Quaternary deposits and 1.35 and 0.3 for streambed sediment, respectively.*

- L179: Were the particles weighted somehow? In the following particle statistics, what does one particle stand for? A certain fraction of groundwater volume? A certain area/volume of bedrock? Please indicate why your choice is the best choice for the research question.

Thanks for the question. Each particle applied in the particle tracing represents a specific cross-sectional area of the release plane; i.e., the CDF is weighted by area. In this study, the groundwater analysis was conducted in the steady state condition, and thus only the spatial impacts could be evaluated. Therefore, conducting particle tracing for 10,000 random particles uniformly spaced over a flat surface could provide us with details of the distribution of deep groundwater, regardless of their location at 500 m depth. The CDFs could be easily transformed into flow-weighted CDFs, since the velocity for each starting position is known, but we feel that this information is irrelevant to the paper. However, we do agree that the definitions of the CDF type should be clear, and thus, we add the following sentence to the paper:

*A particle tracing routine was implemented in the catchment-scale model to analyze the distribution of deep groundwater flow paths (Genel et al., 2013). Each particle applied in particle tracing represents a specific cross-sectional area in the release plane.*

- L196: is "c" the same coefficient as "delta" in line 147?

Not exactly. The "c" in Equation 5 is "$1/\delta$" in Equation 4.

- L213: Why did you use local regions for downscaling? Couldn't it also be a 100x100m region from somewhere else in the world? Or do you assume that there is a correlation between the catchment topography and its streambed-topograpy? I don't think that Wörman, et al., 2007 proved a local correlation between topographies. I think it should be clarified for the reader if a local correlation is assumed or if the regional topography is simply used as a sophisticated random field generator and the topography data could also be taken from somewhere else.

We agree that any region in the world could be used to represent the power spectral properties of topography, but rescaling topography in the real-world domain involves more constraints than just the "power" of the topography function. It has been determined that both streambeds and landscapes are fractal, implying a similarity in geometrical scaling across wavelengths differing by orders of magnitude. Specifically, the variance in landscape elevation shows this similarity over a wide range of wavelengths, but not necessarily other shape measures. Therefore, the method of rescaling streambeds applied in this paper is consistent with the known fractal distributions from the streambed scale to the continental landscape scale (Hino, 1968, Nikora, 1997; Turcotte, 1997; Wörman et al., 2007); however, it also assumes a similarity in the actual (real) shape, which we assume is only a regional behavior trait. This procedure is not supported by the scientific literature, but is used primarily to represent the hyporheic flux and for graphical demonstrations. It has been shown at the streambed interface that the spatially average flux is directly described by the power spectral density of the topography (Morén, et al. 2017, Eq. (7a)), meaning the rescaling of landscapes is not important for the average flux. The following discussion will

be added to the revised paper to address the correlation between hyporheic and regional groundwater hydraulic heads.

*The stream bedform variations governing hyporheic flows were represented by randomly selecting 100 × 100 m² areas from the entire catchment, which were then rescaled and used in Monte Carlo simulations. The rescaling of streambeds applied in this paper is consistent with the known fractal distributions between streambed and continental landscape scales (Hino, 1968; Nikora; 1997; Turcotte, 1997; Wörman et al., 2007); however, it also assumes a similarity in the actual (real) shape, which we assume is only a regional behavior trait. Even though this procedure is not supported by previous studies, it has been shown that the spatially average flux follows the power spectral density of the topography (Morén, et al. 2017, Eq. (7a)), meaning it is not important for the average flux, but it is used to demonstrate the nature of the hyporheic flow fields.*

- L179/246: Both particle tracings should be described in a single chapter.

Both particle tracing will be presented in section 2.6 of the revised paper.

- L248: If intermediate particle traces are those that did not enter the bedrock and deep particles are those that started at 500m depth, you miss the flow that enters the bedrock but not to a depth of 500m. Superimposing two particle tracings with different seeds and without weighting them properly against each other adds some randomness to the results.

Thanks for the insightful comment, which is a correct observation. This section should have been better described to avoid any confusion. The $5\times5\times5$ m³ cubes extracted from the catchment-scale model contained groundwater flow from various spatial scales (including flow that entered the bedrock but did not reach the 500 m depth). However, among all the flows (from different spatial scales), we took the intermediate and deep groundwater (i.e., 500 m depth) flow trajectories for further analyses (i.e., assessing the impact of hyporheic flow fields on deep and intermediate groundwater flows). For this purpose, backward particle tracing was conducted to evaluate the intermediate groundwater flow (i.e., flow particle that did not enter the bedrock). The following new section is added to the paper to describe different flow types:

*Particle tracking was used to identify streamlines that follow hyporheic, intermediate and deep groundwater flows. Deep groundwater was defined as flow that entered the bedrock, and intermediate groundwater was determined to only be present in Quaternary deposits. Hyporheic flow contains streamlines starting and ending in the stream bottom within a 5 m spatial scale.*

In addition, L248 will be revised, as follows:

*Cubes of $5\times5\times5$ m³ at the deep groundwater discharge zones contained groundwater flow from different spatial scales (i.e., from shallow to deep groundwater flows). In addition to the already-evaluated deep groundwater flow, "intermediate groundwater flow" was identified using particle trajectories as starting in recharge areas but confined to Quaternary deposits and sediment layers (i.e., without entering the bedrock domain).*

- L260/Fig2/L179: The release plane for particles was described as "approximately 500 m below the minimum topography elevation". That describes a flat plane. However, the release surface in Fig.2 two shows a curvy plane. Which one is correct and why is the depth "approximately"?

Thanks for the comment. The minimum topography elevation was +117 m a.s.l., and we released the particles from a flat surface located at -380 m a.s.l., which is approximate 500 m from the lowest point of topographical elevation (the exact value is 497 m from this point; we rounded the number for no specific reason). The particles were released from a flat plane (the straight line in Figure 2). We will replace the "curvy plane" in Figure 2 with a flat plane.

- Fig 3 and Fig 6 and the corresponding text could also be placed in the methods section.

Both Figures 3 and 6 present results, and are thus most suited to the Results section. Figure 3 shows the results concerning the impact of topography DEM resolution on the groundwater flow field (the method was already described in lines 161-174), and Figure 6 compares the relative contributions of hydrostatic and dynamic hyporheic

head components to the total hyporheic hydraulic head using Monte Carlo simulation (sections 2.4 and 2.5 describe the method for this).

- Fig 7 and corresponding text: How many particle traces from the deep fraction entered the 5x5m domain to be superimposed by hyporheic flux? According to Fig4 it could only be a handful. Why are the dashed lines so smooth and the solid lines are not? Do they represent the same amount of particle traces?

In total, 1552 deep groundwater particles reached the topographical surface within the catchment area (refer to L295 of the paper), and are represented as dots in Fig. 4. The travel time of each deep groundwater particle was evaluated in a $5{\times}5{\times}5$ m$^3$ area in the absence and presence of hyporheic fluxes, respectively. Then, the evaluated travel times across the sediment layer of all 1552 of the deep groundwater particles were presented in the form of a CDF (cumulative distribution function) plot. Therefore, Figure 7 presents the travel time of all the deep groundwater particles that reached the topographical surface within the catchment area (i.e., 1552). In addition, the dashed and solid lines in Figure 7 represent the same amount of particles. Both the deep and intermediate groundwater flow travel times through the streambed sediment were reduced due to the hyporheic fluxes, but this impact was not uniform for all the particles. Therefore, the range of travel times decreased (dashed lines), which led to smoother lines compared to when the hyporheic flow field was omitted (solid lines).

- L377-400: To be honest, I did not understand what you did here. What is on the y-axis of the CDF? The whole topic "coherent area" needs a better explanation. Why did you choose coherence as a measure? What environmental process would coherence be important for? My interpretation is that you created a list of coherent (however coherence was defined precisely) upwelling patches and calculated their surface area. Now you found, for example, that 50% of these patches had surface areas > 400m². Is that correct? If so, I would strongly recommend not to use a CDF for presentation but a PDF. Nothing is cumulating here, CDFs are easily interpreted as "number of particle traces that reached the surface" where 100 means all particles exited the domain or something similar.

The interpretation of the reviewer seems to be correct. The definition of coherent area in the context of this paper was presented on L269-270: "In this study, a coherent area was defined as an area in which the entire flow reflected only the catchment-scale upward groundwater flow". However, we have added this explanation of coherent areas:

> *The fragmentation of the upwelling of groundwater is defined as a shift in the distribution of coherent areas of upwelling groundwater at the streambed interface towards smaller areas. A coherent upwelling area is defined as a set of all adjacent areas in the numerical model with upward flow. Such areas represent streambed areas with upwelling deep groundwater, whereas other areas are not subjected to upwelling deep groundwater.*

Figure 9 shows the distribution of coherent upwelling areas via a CDF plot, which is the statistic integral of the PDF, i.e., both curves would essentially represent the same thing, translated through a mathematical transformation. As an example, if the CDF plot shows 70% (i.e., on the y-axis) for a 5 m$^2$ coherent area (i.e., on the x-axis), this means that 70% of all the coherent areas are less than 5 m$^2$, which could not be read directly in a PDF plot (however, a PDF plot might have other advantages).

The temperature and chemical composition of groundwater and hyporheic fluxes are different (Kalbus et al., 2009), thus the fragmentation of groundwater coherent areas indicates a fragmentation in aquatic chemistry and temperature, which determines the distribution of aerobic and anaerobic conditions, as well as the biological activities within the streambed sediment (Harvey et al., 2013; Marzadri et al., 2011; Zertnetske et al., 2011). Hence, the degree of fragmentation in the coherent upwelling groundwater plays an important role in the establishment and mitigation of biological communities.

- L409: Why does the scenario assume accumulation? Shouldn't it be steady state in- and outflow at some point?

Yes, this is true. We meant "long-term accumulation and transport" scenario. The sentence will be revised in the paper as follows:

> *One scenario implies that radioactivity accumulates over a long time period in shallow aquatic sediments, which may be incorporated into agricultural products after future land use changes or by groundwater withdrawal used for irrigation purposes.*

- L415: I'm not familiar with dose assessment but if it is based on the idea that groundwater upwelling happens on a large area without fragmentation it is obviously oversimplifying groundwater flow. If so, you should explain the dose model in more detail and propose an improvement to the model.

To our knowledge, there is no previous study that has investigated the impact of hyporheic flux on deep upwelling groundwater in a discharge zone (the factors that affect the sizes of the upwelling areas as well as the travel time of the deep groundwater flow near the bed surface). This neglected factor (induced by the hyporheic flow field) has implications for radionuclide safety assessment near the bed surface. Therefore, we suggested the often-neglected hyporheic impact should be considered in dose assessment models.

**References**

Bhaskar, A. S., Harvey, J. W., and Henry, E. J.: Resolving hyporheic and groundwater components of streambed water flux using heat as a tracer, Water Resources Research, 48, 2012

Boano, F., Revelli, R., and Ridolfi, L.: Reduction of the hyporheic zone volume due to the stream-aquifer interaction, Geophysical Research Letters, 35, 2008.

Elliott, A. H. and Brooks, N. H.: Transfer of nonsorbing solutes to a streambed with bed forms: Theory, Water Resources Research, 33, 123-136, 1997a.

Fox, A., Laube, G., Schmidt, C., Fleckenstein, J., and Arnon, S.: The effect of losing and gaining flow conditions on hyporheic exchange in heterogeneous streambeds, Water Resources Research, 52, 7460-7477, 2016.

Gleeson, T., Marklund, L., Smith, L., and Manning, A. H.: Classifying the water table at regional to continental scales, Geophysical Research Letters, 38, 2011.

Genel, S., Vogelsberger, M., Nelson, D., Sijacki, D., Springel, V., and Hernquist, L.: Following the flow: tracer particles in astrophysical fluid simulations, Monthly Notices of the Royal Astronomical Society, 435, 1426-1442, 2013.

Haitjema, H. M. and Mitchell-Bruker, S.: Are water tables a subdued replica of the topography?, Groundwater, 43, 781-786, 2005.

Harvey, J. W., Böhlke, J. K., Voytek, M. A., Scott, D., and Tobias, C. R.: Hyporheic zone denitrification: Controls on effective reaction depth and contribution to whole-stream mass balance, Water Resources Research, 49, 6298-6316, 2013.

Hino, M.: Equilibrium-range spectra of sand waves formed by flowing water, Journal of Fluid Mechanics, 34, 565-573, 1968.

Jakubick, A.: Analysis of Pu-Release Consequences on the Environmental Geochemistry, in: Scientific Basis for Nuclear Waste Management, Springer, 427-434, 1979.

Kalbus, E., Schmidt, C., Molson, J., Reinstorf, F., and Schirmer, M.: Influence of aquifer and streambed heterogeneity on the distribution of groundwater discharge, Hydrology and Earth System Sciences, 13, 69-77, 2009

Marklund, L. and Wörman, A.: The use of spectral analysis-based exact solutions to characterize topography-controlled groundwater flow, Hydrogeology Journal, 19, 1531-1543, 2011.

Marzadri, A., Tonina, D., and Bellin, A.: A semianalytical three-dimensional process-based model for hyporheic nitrogen dynamics in gravel bed rivers, Water Resources Research, 47, 2011.

Mojarrad, B. B., Riml, J., Wörman, A., and Laudon, H.: Fragmentation of the hyporheic zone due to regional groundwater circulation, Water resources research, 55, 1242-1262, 2019

Morén, I., Wörman, A., and Riml, J.: Design of remediation actions for nutrient mitigation in the hyporheic zone, Water Resources Research, 53, 8872-8899, 2017.

Neretnieks, I.: Analysis of some tracer runs in granite rock using a fissure model, in: Scientific Basis for Nuclear Waste Management, Springer, 411-415, 1979.

Nikora, V. I., Sukhodolov, A. N., and Rowinski, P. M.: Statistical sand wave dynamics in one-directional water flows, Journal of Fluid Mechanics, 351, 17-39, 1997

Ryan, R. J. and Boufadel, M. C.: Evaluation of streambed hydraulic conductivity heterogeneity in an urban watershed, Stochastic Environmental Research and Risk Assessment, 21, 309-316, 2007.

Singh, A., Phogat, V., Dahiya, R., and Batra, S.: Impact of long-term zero till wheat on soil physical properties and wheat productivity under rice–wheat cropping system, Soil and Tillage Research, 140, 98-105, 2014.

Song, J., Chen, X., Cheng, C., Summerside, S., and Wen, F.: Effects of hyporheic processes on streambed vertical hydraulic conductivity in three rivers of Nebraska, Geophysical Research Letters, 34, 2007.

Trauth, N., Schmidt, C., Vieweg, M., Maier, U., and Fleckenstein, J. H.: Hyporheic transport and biogeochemical reactions in pool-riffle systems under varying ambient groundwater flow conditions, Journal of Geophysical Research: Biogeosciences, 119, 910-928, 2014.

Turcotte, D.: Fractals and chaos in geology and geophysics Cambridge University, Press, New York, 1997.

Wang, C., Gomez-Velez, J. D., and Wilson, J. L.: The importance of capturing topographic features for modeling groundwater flow and transport in mountainous watersheds, Water Resources Research, 54, 10,313-310,338, 2018.

Wörman, A., Packman, A. I., Marklund, L., Harvey, J. W., and Stone, S. H.: Fractal topography and subsurface water flows from fluvial bedforms to the continental shield, Geophysical Research Letters, 34, 2007.

Zarnetske, J. P., Haggerty, R., Wondzell, S. M., and Baker, M. A.: Labile dissolved organic carbon supply limits hyporheic denitrification, Journal of Geophysical Research: Biogeosciences, 116, 2011.